# Decoding a cryptic mechanism of metronidazole resistance among globally disseminated fluoroquinolone-resistant *Clostridioides difficile*

Abiola O. Olaitan [1,2,11], Chetna Dureja[1,11], Madison A. Youngblom [3], Madeline A. Topf[3], Wan-Jou Shen[1], Anne J. Gonzales-Luna[4], Aditi Deshpande [1], Kirk E. Hevener[5], Jane Freeman[6,7], Mark H. Wilcox [6,7], Kelli L. Palmer[8], Kevin W. Garey [4], Caitlin S. Pepperell [9,10,12] ✉ & Julian G. Hurdle[1,12] ✉

Severe outbreaks and deaths have been linked to the emergence and global spread of fluoroquinolone-resistant *Clostridioides difficile* over the past two decades. At the same time, metronidazole, a nitro-containing antibiotic, has shown decreasing clinical efficacy in treating *C. difficile* infection (CDI). Most metronidazole-resistant *C. difficile* exhibit an unusual resistance phenotype that can only be detected in susceptibility tests using molecularly intact heme. Here, we describe the mechanism underlying this trait. We find that most metronidazole-resistant *C. difficile* strains carry a T-to-G mutation (which we term P*nimB^G*) in the promoter of gene *nimB*, resulting in constitutive transcription. Silencing or deleting *nimB* eliminates metronidazole resistance. NimB is related to Nim proteins that are known to confer resistance to nitroimidazoles. We show that NimB is a heme-dependent flavin enzyme that degrades nitroimidazoles to amines lacking antimicrobial activity. Furthermore, occurrence of the P*nimB^G* mutation is associated with a Thr82Ile substitution in DNA gyrase that confers fluoroquinolone resistance in epidemic strains. Our findings suggest that the pandemic of fluoroquinolone-resistant *C. difficile* occurring over the past few decades has also been characterized by widespread resistance to metronidazole.

*Clostridioides difficile* infection (CDI), a leading cause of hospital-associated diarrhea, has attracted international attention due to worsening clinical outcomes resulting from the global spread of epidemic strains of PCR ribotype (RT) 027[1–3]; RT027 belongs to phylogenetic Clade 2 (Supplementary Data 1 shows the relationship between ribotype and phylogenetic classifications based on genomes of strains; in this study, strains are classified based on their ribotype and/or phylogenetic clade, unless otherwise specified)[4,5]. These strains caused outbreaks of CDI across North America, UK, Europe and Latin America with an increased incidence of severe illness, morbidity, and mortality[1,2,6]. These global outbreaks also demarcate the pandemic era of CDI[3]. During this era metronidazole and vancomycin were the main two antibiotics used to treat CDI, until fidaxomicin was approved by the FDA in 2011. However, due to declining efficacy, metronidazole is no longer recommended as a first-line drug for adult CDI in the updated IDSA/SHEA and ESCMID CDI guidelines[7,8]. This represents a significant shift in the treatment paradigm of CDI[9–12], with metronidazole reserved as an intravenous therapy in combination with

vancomycin for fulminant CDI[7,8]. These changes in CDI therapeutics, due to declining metronidazole efficacy, warrant elucidation of the microbial genetic factors affecting this drug and the global CDI epidemiology. The microbial genetic determinants of metronidazole resistance are poorly understood.

Metronidazole was established as an antibiotic for CDI following clinical trials in the 1980s and 1990s, where it not only showed comparable clinical success rates to vancomycin but was much less expensive[13,14]. However, over the last two decades it has become less effective, compared to vancomycin[15,16]. This was originally seen in a randomized clinical trial, conducted between 1994 and 2002, where metronidazole demonstrated an 84% cure rate compared to 97% seen with vancomycin[15]. In a second clinical study conducted between 2005 and 2007, vancomycin exhibited superior cure rates to metronidazole, 81.1% compared to 72.7%[16]. These two studies reveal that metronidazole became less effective in the epidemic era. Indeed, metronidazole treatment failures in Quebec more than doubled from 9.6% in 1991–2002 to 25.7% in 2003–2004[17], which is also the region that reported the first outbreak of epidemic RT027[2]. The reasons for the reduced clinical utility of metronidazole have been a longstanding mystery. One possibility is that increased use of metronidazole in response to rising rates of CDI imposed selection pressures that enabled the emergence and dissemination of drug-resistant *C. difficile*.

Metronidazole is a nitroimidazole prodrug that is activated in cells in reactions conducted by oxidoreductases, such as pyruvate-ferredoxin/flavodoxin oxidoreductase (PFOR), to form reactive species (e.g., anion, nitroso, and hydroxylamine intermediates) that damage DNA and proteins and deplete cellular thiols[18]. Metronidazole therefore triggers oxidative/nitrosative stress in *C. difficile*[19]. Interestingly, clinical isolates of *C. difficile* are poorly characterized in terms of genetic determinants that confer resistance to metronidazole[18,20]. A high copy number plasmid (pCD-METRO) was recently reported to confer clinical resistance to metronidazole, in a *C. difficile* isolate obtained from a patient failing metronidazole therapy[20]. However, the overwhelming majority of metronidazole-resistant *C. difficile* do not encode pCD-METRO[20]. A survey of 10,330 publicly available genomes found pCD-METRO in just 15 strains, indicating it is exceedingly rare[21]. Further complexity arose from reports of metronidazole-resistant *C. difficile* with unstable minimum inhibitory concentrations (MICs). This has likely resulted in the systematic underestimation of the prevalence of metronidazole-resistant *C. difficile*. Heme was recently shown to be essential for the accurate detection of metronidazole resistance in *C. difficile*, and the majority of metronidazole-resistant *C. difficile* exhibit heme-dependent resistance via an unknown mechanism[22,23]; pCD-METRO does not confer heme-dependent resistance[22]. Due to the discovery of the heme-dependent phenotype, the underlying mechanism of metronidazole resistance, which was previously thought to be unstable, can now be elucidated.

Herein, we describe the discovery and genetic validation of the mechanism for heme-dependent resistance to metronidazole and reveal its association with the global transmission of epidemic *C. difficile* through genome-wide association studies and population genetics. We discovered that epidemic strains evolved a common variant in the regulatory promoter of 5-nitroimidazole reductase (*CDR20291_1308*, annotated as *nimB*), causing its conversion from a cryptic to a constitutively expressed resistance gene. We further show that the protein *C. difficile* NimB (*Cd*NimB) is a heme-binding flavoenzyme that bioreductively inactivates 5-nitroimidazoles to corresponding amines, and that its substrate profile includes 4-nitrobenzoic acid and 2-nitroimidazole. Finally, we discovered a strong correlation between the *nimB*-promoter variant and a Thr82Ile mutation in DNA Gyrase A (GyrA), which confers fluoroquinolone resistance and has been linked to the global spread of epidemic *C. difficile*[24–26]. These co-occurring variants are strongly associated with epidemic strains,

suggesting that they are advantageous in this setting. Therefore, these findings update the current paradigm for the rapid global spread of fluoroquinolone-resistant epidemic *C. difficile* strains that are associated with poor clinical outcomes[24–26], also indicating an association with resistance to metronidazole.

## Results

### Heme mediates metronidazole resistance

We performed metronidazole susceptibility testing, with and without heme, on *C. difficile* strains of varying ribotypes and Clades (Supplementary Data 1) from the US and Europe ($n = 405$). Forty one percent of the strains displayed a ≥4-fold increase in metronidazole MICs in the presence of heme (MICs = 1–16 μg/ml), compared to those in the absence of heme (MICs = 0.25–1 μg/ml) and this criterion was used to designate strains as showing heme-dependent resistance to metronidazole (Fig. 1a, Supplementary Data 1). We did not detect any metronidazole-resistant isolates that were independent of heme. Conversely, metronidazole-susceptible strains displayed a 2-fold or less difference in MICs, with or without heme. Heme was also required for strains to exhibit MICs above the EUCAST resistance breakpoint of >2 μg/ml; Supplementary Fig. 1a, b).

### Transcriptome analysis confirms heme attenuates cellular toxicity of metronidazole

R20291, a well-known model strain of epidemic RT027 (Clade 2) isolated from a hospital outbreak in 2003–2006 in Stoke Mandeville, UK[27], was adopted as our main research strain since it shows heme-dependent resistance to metronidazole (MICs of 2–4 μg/ml and 0.25–0.5 μg/ml, with and without heme, respectively). Examination of transcriptional responses of R20291 exposed to metronidazole (2 μg/ml) and heme (5 μg/ml) for 30 min showed that heme attenuated the upregulation of pathways associated with metronidazole toxicity. Without heme, there were 285 upregulated genes (≥2-fold) by metronidazole, but only 44 of these genes were upregulated when heme was present (Fig. 1b, c). Pathways induced by metronidazole toxicity that were mitigated by heme include DNA replication and repair (e.g., ribonucleotide reductase, *nrdF*) and stress responsive chaperones (e.g., co-chaperonin, *groES* and bacterial Hsp70 chaperone, *dnaK*). By itself, heme minimally affected *C. difficile* transcriptome (Supplementary Fig. 2a). Validation of the overall quality of the RNA-seq data by qRT-PCR showed a high correlation of $R^2 = 0.94$ between both datasets (Supplementary Fig. 2b, c). We next tested whether heme could also quench metronidazole toxicity in metronidazole-susceptible *C. difficile* CD196 (a pre-epidemic RT027 strain that was isolated in 1985 in Paris). However, heme did not attenuate gene expression associated with metronidazole toxicity in CD196 (Fig. 1d, Supplementary Fig. 2d, e). It was postulated that the heme-binding transcriptional regulator HsmA[28] may be associated with heme-dependent metronidazole resistance, as the gene appears inactivated in RT010 strains[22]. However, we did not find any evidence for the involvement of HsmA or the heme detoxifying system encoded by *hatRT*[29] in mediating heme-dependent metronidazole resistance (Fig. 1e, f). Collectively, the data suggested that other genetic factors mediate heme-dependent metronidazole resistance. The raw RNA-Seq data are deposited in NCBI database under accession number PRJNA880780.

### Identification of *C. difficile* 5-nitroimidazole reductase (*Cd*NimB) as a mechanism for heme-dependent metronidazole resistance

To screen for genetic loss of heme-dependent metronidazole resistance we generated ~7488 transposon (Tn) mutants in R20291 using vector pRPF-215 that carries an ATc-inducible *Himar1*[30]. This identified two Tn mutants that were eightfold more susceptible to metronidazole (MICs of 0.5 μg/ml), compared to the wildtype (MIC of 4 μg/ml). Genome sequencing revealed that the two Tn mutants each had single insertions in *CDR20291_1308* or *CDR20291_2676* at positions 1547478

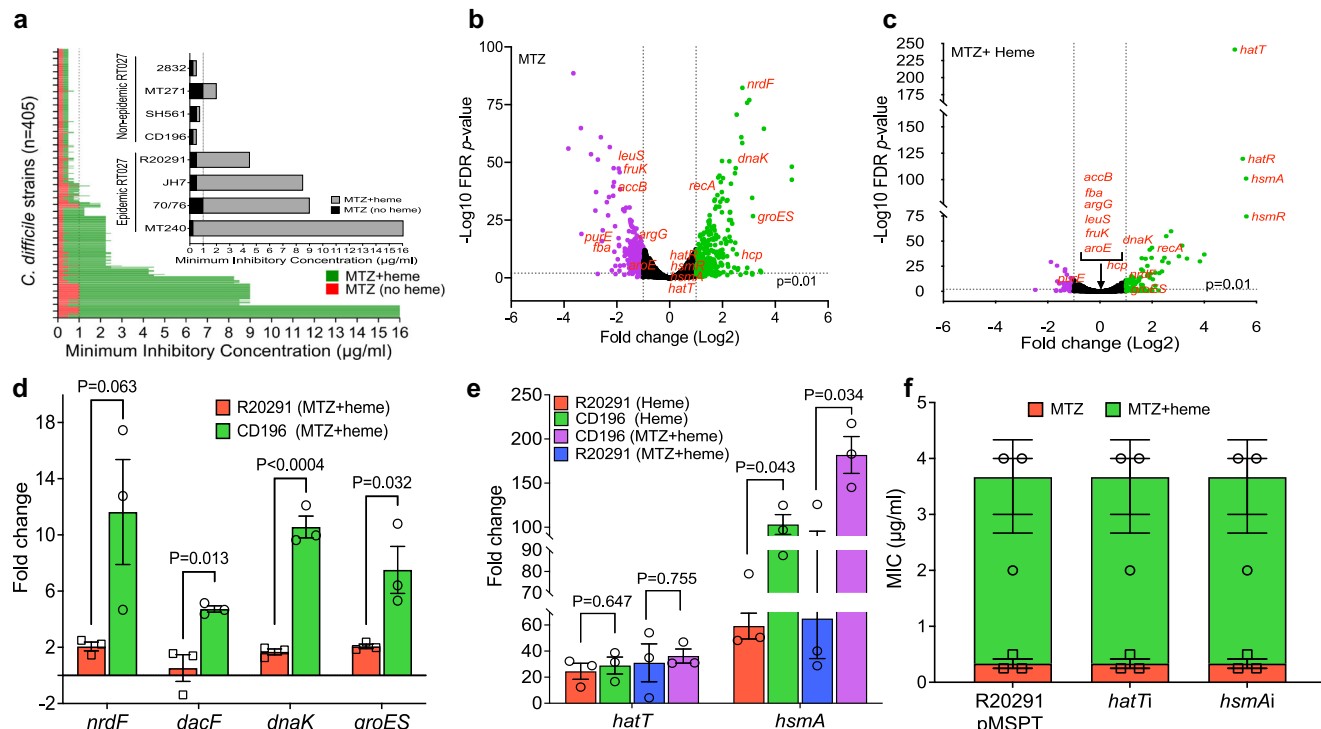

**Fig. 1 | Heme attenuates cellular toxicity of metronidazole (MTZ) in epidemic *C. difficile* strains. a** Minimum inhibitory concentrations (MICs) of MTZ were determined against global clinical isolates of *C. difficile* (*n* = 405) in the presence and absence of heme. Isolates with heme-dependent MTZ resistance exhibited ≥4-fold higher MICs in BHI agars containing heme, compared to agars without heme. An MTZ MIC of 1 μg/ml (dashed line) was found to be associated with poor clinical outcomes[58], although the EUCAST resistance breakpoint is 2 μg/ml. Inset illustrates selected RT027 epidemic and non-epidemic strains that are resistant or susceptible to MTZ in heme, respectively. Non-epidemic RT027 were defined as previously described[24] (i.e., lacking fluoroquinolone resistance SNPs in gyrase A and large transposons found in epidemic RT027). MICs were from two biological replicates with two technical replicates. Transcriptome response of epidemic R20291 (OD600 nm ≈ 0.2) exposed for 30 min to MTZ (2 μg/ml) in the absence (**b**) and presence (**c**) of heme (5 μg/ml). The volcano plots indicate differentially expressed genes and their statistical significance; the purple and green dots indicate significantly downregulated and upregulated genes, respectively; red-highlighted genes are cell stress-responsive and metabolic genes whose expression were altered by MTZ but were attenuated by heme. The RNA-seq is based on two biological replicates, as the third replicate was rejected in FastQC due to poor quality. Pearson correlation of $R^2$ = 0.94 in Supplementary Fig. 2b shows RNA-seq transcriptional changes were validated by qPCR (three biological replicates). In (**b**) and

(**c**), differential gene expression was determined with edgeR, an RNA-seq analysis tool with default statistical analysis of false discovery rate (FDR) ≤ 0.01 and Log₂ fold change set to ≥1; output *p* values for each gene were converted to negative logarithm (−Log10) and plotted against Log2-fold change in Graphpad prism 9.4.1. **d** Transcriptional response of epidemic R20291 and non-epidemic CD196 to MTZ (2 μg/ml) in the presence of heme (5 μg/ml). qPCR analysis of transcription patterns of genes indicative of MTZ toxicity demonstrates heme is not protective for non-epidemic CD196. **e** Transcriptional analysis of heme sensing/detoxifying genes *hatT* and *hsmA* in heme with or without MTZ, indicate that CD196 and R20291 both highly express these genes, as determined by qPCR. Data from (**d**) and (**e**) were statistically analyzed in Graphpad prism 9.4.1 by two-tailed multiple unpaired *t* test with correction for multiple comparisons using the Holm-Šídák method and alpha set to 0.05; data in each plot were from three biological replicates. **f** Heme sensing and detoxifying systems do not mediate heme-dependent MTZ resistance. In R20291, silencing *hatT* or *hsmA* had no effect on heme-dependent resistance, indicating that these genes are unlikely to contribute to heme-dependent resistance; anti-sense RNA to *hatT* or *hsmA* was cloned into vector pMSPT and induced by 64 ng/ml of anhydrotetracycline; qPCR indicated *hatT* mRNA decreased by ~7-fold (i.e., −7.21 ± 1.95) and *hsmA* by -18-fold (i.e., −18.46 ± 10.66). Data in (**d**), (**e**), and (**f**) are plotted as mean ± standard error of mean.

(CDS position 464 of 468 nt) and 3152802 (CDS position 1151 of 2412 nt), respectively in the R20291 genome (FN545816.1). The two genes are annotated as 5-nitroimidazole reductase (*nimB*) and cysteine protease (*cwp84*), respectively. Cwp84 is responsible for cleaving the surface layer (S-layer) protein precursor SlpA into low- and high-molecular weight subunits that form the paracrystalline S-layer. Although *cwp*84 is a non-essential gene, deletion mutants have slower growth rates, which might impose pleiotropic effects that increase metronidazole susceptibility of the *cwp84* Tn mutant[31]. *C. difficile* NimB (*Cd*NimB) belongs to the pyridoxamine 5′-phosphate oxidase family of proteins and is related to NimA protein from *Terrisporobacter* spp. WP_228108130.1 (60% identity) and NimB from *Bacteroides fragilis* WP_063854490.1 (45% identity); noteworthy, several isoforms of Nim proteins exist (e.g., NimA-L) and may be encoded chromosomal or plasmid encoded[32–34]. Nim proteins are thought to cause resistance to nitroimidazoles by rapidly converting their nitro groups to antimicrobial inactive aminoimidazole derivatives, thereby avoiding the generation of reactive species[35]. It is unclear, however, to what extent

chromosomally encoded *nimB* confers resistance to metronidazole in *C. difficile* since this gene appears ubiquitous in this specie and its mere presence does not directly result in resistance. Despite this, we chose to further study *nimB* because of the marked phenotype of the Tn mutant (R20291-Tn::*nimB*) and the high likelihood that in *C. difficile nimB* may be another cryptic resistance mechanism[36]. Hence, when we generated a *nimB* knockout mutant by allelic exchange in R20291, the mutant recapitulated the metronidazole-susceptible phenotype of R20291-Tn::*nimB* (i.e., MIC = 0.25 μg/ml for R20291Δ*nimB* vs. 4–8 μg/ml for the wildtype in the presence of heme, Fig. 2a). Noteworthy, R20291Δ*nimB* was slightly more susceptible to metronidazole than R20291-Tn::*nimB*; perhaps this is a result of the *ermB* insertion occurring just after the codon for the C-terminal arginine-154, so *Cd*NimB of R20291-Tn::*nimB* may have some activity, albeit insufficient to cause resistance to wildtype levels. Importantly, heme-dependent resistance was restored (MICs=4-8 μg/ml) upon complementation of R20291Δ*nimB* and R20291-Tn::*nimB* with plasmid-borne wildtype *nimB* from R20291 (Fig. 2a).

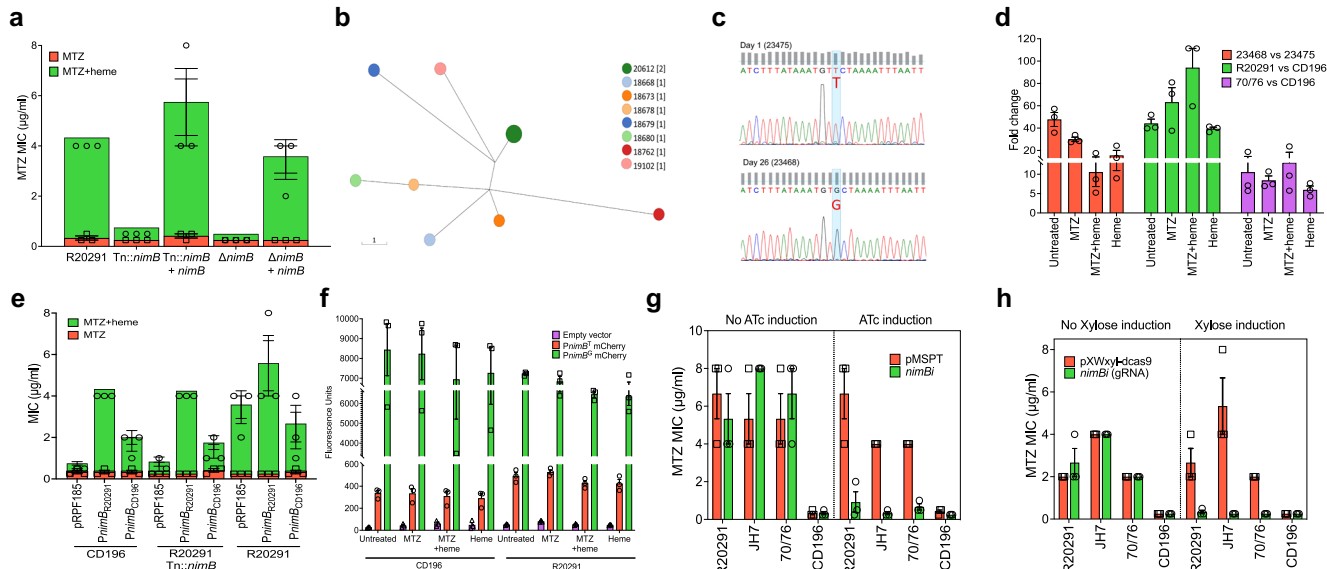

**Fig. 2 | Constitutive transcription of *nimB* is associated with metronidazole (MTZ) resistance and results from a mutation in the −10 promoter. a** Screening of a transposon (Tn) library identified that insertional inactivation of *nimB* abrogated resistance in R20291. The allelic deletion of *nimB* confirmed its role in resistance; in both mutants, resistance was restored by complementation with wildtype *nimB*, expressed from its native promoter. **b** Sequential isolates from a patient failing MTZ therapy were collected on day 1 (before therapy) and on day 26 (after therapy ended); the day 1 isolate (23475, MIC = 0.25 µg/ml) and day 26 isolate (23468, MIC = 1.0 µg/ml) were MTZ-susceptible and -resistant, respectively. In Enterobase, core-genome multilocus sequence typing showed the two isolates had the same hierarchical clustering (HC0), indicating they have identical core genomes or at least do not differ by more than two SNPs. cgMLST Ninja Neighbor Joining GrapeTree of *C. difficile* belonging to ST350; 23475 and 23468 clustered in the same node. The scale bar indicates the number of allelic differences. **c** Sanger sequencing showed 23468 has a T to G mutation in the −10 *nimB*-promoter when compared to 23475, confirming a whole genome comparison; Sanger sequencing was done on a separate colony (i.e., *n* = 1 per strain) to that used for genome sequencing; this mutation is referred to as P*nimB^G* in 23468, while the wildtype is

designated P*nimB^T* in 23475. **d** qPCR analysis of *nimB* transcription in resistant and susceptible strains carrying P*nimB^G* and P*nimB^T*, respectively. Resistant strains (23468, R20291, and 70/76) had higher levels of *nimB* mRNA than susceptible strains (23475 and CD196); *nimB* transcription was constitutive in resistant strains. Strains were cultured in various conditions (MTZ [2 µg/ml] and heme [5 µg/ml]); shown are fold mRNA amounts in resistant strains relative to susceptible strains. **e** Relationship between resistance and variations in *nimB* between non-epidemic CD196 and epidemic R20291. The *nimBs* of CD196 and R20291 were expressed from their native promoters in the indicated strains. **f** Comparison of the promoter strengths of P*nimB^G* and P*nimB^T* based on the transcription and fluorescence of mCherryOpt. Fluorescence was normalized to OD$_{600}$nm culture density. P*nimB^G* was associated with constitutive expression, reflected by higher fluorescence. Genetic silencing of *nimB* reverses resistance. *nimB* was silenced by an antisense RNA (asRNA) (**g**) and by CRISPR interference with two guide RNAs (gRNAs) (**h**). AsRNA was induced by anhydrotetracycline (64 ng/ml) from P*tet* promoter in vector pMSPT, while gRNA was induced by xylose (1% w/v) from P*xyl* promoter in vector pXWxyl-dcas9. Data in (**a**), (**d**), (**e**), (**f**), (**g**) and (**h**) are shown as the mean ± standard error of mean from three biological replicates.

## Heme-dependent resistance results from a SNP in *nimB* −10 promoter

To identify genetic mechanisms in clinical strains we sequenced a pair of isolates obtained from a single patient, before and after therapy with metronidazole; these isolates were from the MODIFY clinical trials of bezlotoxumab[37]. The metronidazole-susceptible isolate (23475, RT014) was recovered at baseline diagnosis and had an MIC of 0.25 µg/ml, while the second isolate (23468, RT014) obtained after metronidazole therapy (day 26 after diagnosis) had an MIC of 1 µg/ml (no other patient information was disclosed). To assess genetic relatedness, the genomes of 23475 and 23468 were analyzed by core-genome multilocus sequence typing (cgMLST) in EnteroBase, which indicated that they formed a unique HC0 hierarchical cluster that is indicative of an indistinguishable core-genome sequence type (both were designated as cgMLST 20612; Fig. 2b). Genome alignment and Sanger sequencing revealed that the resistant strain 23468 evolved a single nucleotide polymorphism (SNP) of T to G in the predicted −10 regulatory promoter region (i.e., TTCTAAAAT to TGCTAAAAT) (Fig. 2c). The SNP differences were designated as P*nimB^T* and P*nimB^G*, for the susceptible wildtype and resistant mutant, respectively. Because the −10 region influences RNA polymerase activity[38,39], we tested whether the SNP affected *nimB* transcription. qRT-PCR revealed 23468 had on average 16- to 40-fold greater amounts of *nimB* transcripts compared to 23475 (Fig. 2d). Elevated *nimB* transcription in 23468 was not dependent on heme or the presence of metronidazole, suggesting the gene was constitutively transcribed. Interestingly, 23468 and 23475 exhibited

comparable growth rates in BHI with or without heme, suggesting that constitutive expression of *nimB* did not affect fitness (Supplementary Fig. 3).

Constitutive transcription may also explain why differential expression of *nimB* was not seen in our RNA-seq. We next compared the regulatory regions of *nimB* in pre-epidemic CD196 (susceptible) with epidemic R20291 (resistant). This revealed that CD196 encoded P*nimB^T*, whereas R20291 encoded P*nimB^G*, suggesting that evolution of a SNP in the regulatory region of *nimB* engendered metronidazole resistance. We therefore compared *nimB* mRNA levels in different metronidazole-susceptible strains (i.e., CD196 and CD630 carrying P*nimB^T*) and resistant strains (i.e., R20291 and 70/76 carrying P*nimB^G*). R20291 exhibited ~39 ± 1- to 94 ± 17-fold or 81 ± 5- to 203 ± 100-fold greater *nimB* mRNA relative to the susceptible strains (CD196 and CD630), with or without heme or metronidazole (Fig. 2d, Supplementary Fig. 4a). The resistant strain 70/76 also had greater amounts of *nimB* mRNA, compared to susceptible strains (Fig. 2d, Supplementary Fig. 4a). Overall, all resistant strains had lower C$_T$ values, indicative of elevated *nimB* mRNA when compared to susceptible strains (Supplementary Fig. 4b).

## P*nimB^G* enhances transcription of *nimB*

Gene expression from P*nimB^T* and P*nimB^G* was examined in several ways to establish why *nimB* mRNA levels differ. Firstly, *nimB* from CD196 or R20291 were ectopically expressed under their cognate promoters in susceptible CD196 and R20291-Tn::*nimB*. In susceptible

backgrounds, the P$nimB^T$ nimB sequence from CD196 conferred lower MICs (1–2 µg/ml) than P$nimB^G$ from R20291 (MICs of 4–8 µg/ml) (Fig. 2e). Secondly, because NimB of CD196 and R20291 differ by a single Leu155Ile polymorphism, we examined the effect of this difference by expressing both genes under the tetracycline-inducible promoter (P$tet$). Under P$tet$, both polymorphisms conferred equivalent heme-dependent resistance in R20291-Tn::nimB (Supplementary Fig. 4c), contradicting a previous hypothesis that Leu155Ile influences metronidazole resistance[40]. Thirdly, we used the mCherryOpt reporter[41] to compare the promoter strengths of P$nimB^T$ and P$nimB^G$ in different genetic backgrounds, under drug-free and metronidazole conditions. As shown in Fig. 2f, CD196 expressing P$nimB^G$::mCherryOpt had higher fluorescence (6943 ± 1729 to 8437 ± 1308 RFU) when compared to P$nimB^T$::mCherryOpt (290 ± 43 to 335 ± 26 RFU). Similarly, R20291 expressing P$nimB^G$::mCherryOpt displayed higher fluorescence than P$nimB^T$::mCherryOpt (6354 ± 454 to 7219 ± 56 RFU and 423 ± 38 to 527 ± 20 RFU, respectively). These trends were consistent in other susceptible and resistant strains tested i.e., P$nimB^G$::mCherryOpt ranged from 2932 ± 508 to 5222 ± 492 RFU and P$nimB^T$::mCherryOpt from 268 ± 18 to 436 ± 11 RFU (Supplementary Fig. 4d, e); qPCR confirmed that the higher mCherry fluorescence from P$nimB^G$ corresponded with greater amounts of mCherry mRNA, compared to P$nimB^T$ (Supplementary Fig. 4f). Fourthly, since nimB is constitutively transcribed under P$nimB^G$, we reasoned genetic silencing would restore susceptibility to metronidazole. We therefore silenced nimB with two different methods[42], with vector pMSPT, considered to block mRNA translation by antisense RNA (asRNA) targeting the UTR region, or by Crispr-interference to block transcription with a guide RNA (gRNA) to nimB expressed from vector pXWxyl-dcas9[42]. Expression of asRNA to nimB from pMSPT reversed heme-dependent resistance in R20291 and other RT027 strains tested, compared to the empty vector or uninduced controls (Fig. 2g). Silencing nimB with gRNA also increased the susceptibility of these strains to metronidazole, when compared to the empty vector or uninduced controls (Fig. 2h). Taken together, these data show heme-dependent metronidazole resistance involved constitutive transcription of nimB, which is driven by the stronger promoter P$nimB^G$.

## C. difficile NimB (CdNimB) is a heme-binding protein

We next revealed that NimB is a hemoprotein, explaining why heme is required for metronidazole resistance. There are no reports that Nim proteins bind heme. However, Phyre2-predicted the CdNimB was structurally related to the heme binding flavocytochromes Anf3 nitrogenase from *Azotobacter vinelandii* (PDB ID 6RK0) and MSMEG_4975 flavin/deazaflavin oxidoreductases from *Mycobacterium smegmatis* (PDB ID 4YBN)[43,44]. Anf3 and MSMEG_4975 are pyridoxamine 5′-phosphate oxidase family proteins that bind heme through the structurally homologous proximal histidine-70 and histidine-62, respectively[43,44]. To discover the heme binding motif(s) in NimB proteins, we first aligned the *B. thetaiotaomicron* NimB crystal structure (PDB ID 2FG9) with that of Anf3, revealing that BtNimB histidine-50 was homologous to Anf3 histidine-70 (Supplementary Fig. 5). Similarly, sequence alignment and structural homology modeling indicated that histidine-55 of CdNimB was homologous to Anf3 histidine-70 (Fig. 3a, b). Interestingly, the amino acid is also identical to histidine-71 in *Deinococcus radiodurans* (DrNimA, Q9RW27; Fig. 3a), which was hypothesized to bind an unidentified cofactor that affected the color of the enzyme in suspension[45]. Hemoquest[46], an algorithm that predicts heme-binding motifs, also identified CdNimB histidine-55 as a candidate heme binding residue, along with cysteine-56, histidine-61, cysteine-101, and tyrosine-119. Therefore, to define the crucial residue, NimB variants (His55Ala, Cys56Ala, His61Ala, and Tyr119Ala) were created by site-directed mutagenesis, expressed in susceptible R20291-Tn::nimB and tested for loss of resistance. The Ala-55 mutant was the only variation that

did not cause heme-dependent resistance (Fig. 3c, Supplementary Fig. 6a).

Spectrophotometric analyses of the Soret band, which signifies heme binding to proteins, showed that recombinant wildtype CdNimB, from R20291, was almost twice as effective at binding heme than the Ala-55 mutant. Incubation of wildtype CdNimB (10 µM) with increasing amounts of heme (1–7 µM) produced a Soret band at 416 nm (Fig. 3d), confirming the development of a heme–protein complex. Increasing concentrations of heme caused higher peak intensity, reaching saturation at ≥4 µM of heme (Fig. 3d). In contrast, equimolar NimB-Ala-55 displayed at least 50% lower Soret band intensities (Fig. 3e), indicating reduced formation of the protein–heme complex. This was confirmed visually, as enzyme-heme suspensions from the wildtype CdNimB-heme suspensions were more intensely reddish-brown color, indicating heme-protein complexation (Supplementary Fig. 7a). Considering 10 µM of CdNimB was saturated by 4 µM of heme, which is less than the protein concentration, we reasoned that a small fraction of the protein contained cellularly derived heme; indeed, a proportion of hemoproteins expressed in *E. coli* BL21(DE3) are purified in the holo-state[47]. Using a peroxidase colorimetric assay, to detect low concentrations of heme (i.e., based on hemin), we found that the wildtype CdNimB contained an estimated 1.4–1.7-fold more heme than the Ala-55 mutant (Supplementary Fig. 7b). Thus, in an intracellular environment wildtype CdNimB appears to bind heme more readily than the Ala-55 mutant. Interestingly, pyruvate was previously thought to be a cofactor for DrNimA[48], but this was later disproved in a study[45] reporting that the purified DrNimA showed a Soret band, which was absent in its cognate His-71 mutant (homologous to His-55 in CdNimB, Fig. 3a)[45]; this is consistent with our findings that CdNimB binds heme and shows a Soret band, which is reduced in the His-55 mutant. Taken together, CdNimB is a heme-binding protein, with histidine-55 most likely acting as its proximal ligand, which is a characteristic expected to be conserved in other Nim proteins.

## NimB inactivates nitroaromatic compounds in a heme-dependent manner

Mechanistically, Nim proteins are thought to reduce the nitro-group of 5-nitroimidazoles to their non-toxic amino derivatives, following the direct transfer of electrons from unknown cofactor(s)[35,45,49]. In support of Nim inactivating nitroimidazoles, it was reported that *B. fragilis* cells carrying NimA reduced dimetridazole to amino derivatives in Wilkins–Chalgren medium, which contain heme[35]. To test the nitroreductase activity of CdNimB under anaerobic conditions, we adapted a nitroreductase assay[43], in which oxidation of NADPH (electron donor) is coupled to reduction of FAD and heme. The Bratton–Marshall reagent that is specific for primary aromatic amines was used to colorimetrically detect the aromatic amino product under anaerobic conditions. Experiments with wildtype CdNimB showed that its nitroreductase activity was dependent on NADPH and a nitro-aromatic substrate (i.e., 4-nitrobenzoic acid, 2-nitroimidazole, metronidazole; Supplementary Fig. 8a, b). The addition of only NADPH and nitrobenzoic acid resulted in enzymatic activity (Supplementary Fig. 8b), since the protein contains endogenous heme and flavin(s) cofactors that were natively bound during expression of the protein in *E. coli* (Supplementary Fig. 7a–c). Unfortunately, attempts to remove heme using methylethylketone[50] were unsuccessful, as the enzyme precipitated (*data not shown*). Nevertheless, the addition of heme (10 µM) and/or FAD (10 µM) significantly increased enzymatic activity (Supplementary Fig. 8b), supporting a role for these cofactors in reducing the nitro-group. We then compared the activities of wildtype CdNimB and the Ala-55 CdNimB mutant using a variety of substrates (4-nitrobenzoic acid, metronidazole, dimetridazole and 2-nitroimidazole), with Bratton-Marshall detection of aromatic amines. With all substrates, wildtype CdNimB displayed significantly higher enzymatic activity than the Ala-55 CdNimB mutant (Fig. 3f).

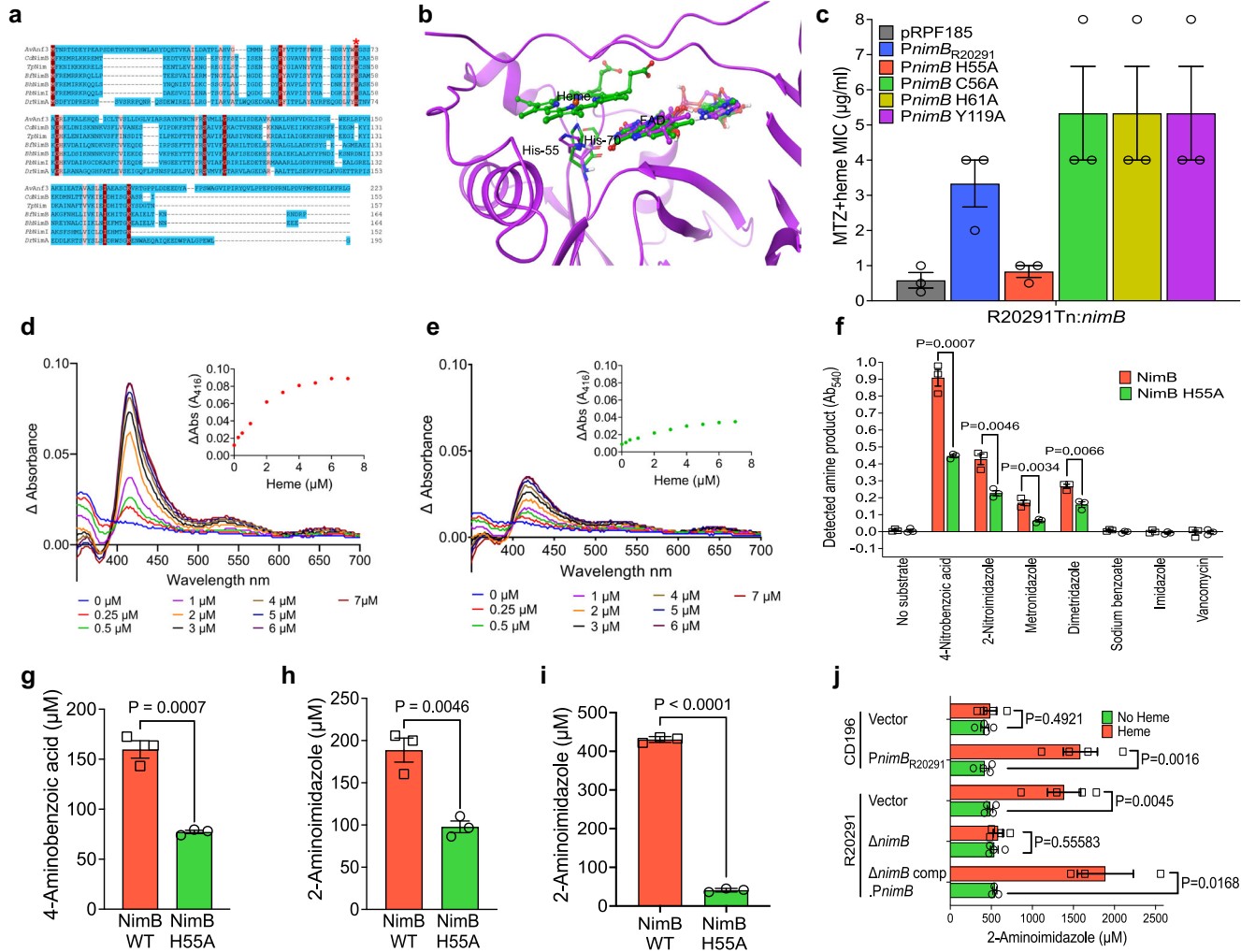

**Fig. 3 | *Cd*NimB is a heme-binding nitroreductase. a** Sequence alignment of Nim proteins from *C. difficile* and other bacteria, with the structurally related heme binding flavocytochrome Anf3 nitrogenase from *Azotobacter vinelandii* (AvAnf3, accession no. 6RK0). Histidine-70 (red asterisk) is the proximal heme-binding ligand of Anf3 and is a conserved amino acid in Nim proteins e.g., histidine-55 in *Cd*NimB. Other sequences and accession numbers are from: *Tp, Terrisporobacter petrolearius* WP_228108130 (https://www.ncbi.nlm.nih.gov/protein/2129663836); *Bf, Bacterioides fragilis* WP_063854490.1; *Bh, Brachyspira hampsonii* WP_039955657; *Dr, Deinococcus radiodurans* Q9RW27; and *Pb, Prevotella baroniae* ACR40098.1. **b** Structural alignment with Anf3 (green ball and stick), a *Cd*NimB homology model (purple ribbons) shows close structural similarity. The *Cd*NimB model was generated from the *B. thetaiotaomicron* NimB X-ray structure bound to FAD (PDB ID 2FG9) and aligned with Anf3, which has both FAD and heme bound (see Supplementary Fig. 5). The model shows *Cd*NimB histidine-55 (purple) is structurally equivalent to histidine-70 (green) of Anf3; the FAD domains are also conserved in the two proteins. FAD and heme of Anf3 are represented as ball/stick with green carbons; the FAD of *Cd*NimB is shown as ball/stick with purple carbons; for simplicity only histidine-70 of the Anf3 protein is shown. **c** Alanine mutagenesis identified histidine-55 mediates heme-dependent metronidazole resistance (MTZ). Among different alanine mutants, only His55Ala mutant did not exhibit resistance when expressed in R20291-Tn::*nimB*; the *nimB* variants were expressed in pRPF185 under P*nimB*G. Absorption spectra showing wildtype *Cd*NimB (**d**) binds heme, but this is attenuated in the His55Ala *Cd*NimB mutant (**e**). The spectra were obtained by adding increasing concentrations of heme (i.e., hemin, 1–7 μM) to 10 μM of protein (data are representative of three experimental replicates). Heme was titrated until saturation was reached, where there were no further significant changes in absorbance readings. The buffer was similarly titrated with heme and values subtracted

from the protein spectral data. The insets show binding saturation curves based on the change in absorbance at 416 nm as a function of heme concentration. **f** Comparison of nitroreductase activities of wildtype *Cd*NimB and Ala-55 mutant. Reduction of various nitroaromatics were tested in reaction containing *Cd*NimB (10 μM), heme (10 μM), FAD (10 μM) and NADPH (3 mM). Reactions were incubated for 2 h, and formation of aromatic amines detected using Bratton-Marshall assay; imidazole, sodium benzoate and vancomycin were non-nitroaromatic negative controls. Corresponding assay development is shown in Supplementary Fig. 8. Data were statistically analyzed in Graphpad prism 9.4.1 by two-tailed multiple unpaired *t* test with correction for multiple comparisons using the Holm-Šídák method and alpha set to 0.05. Comparison of nitroreductase activities of wildtype *Cd*NimB and His55Ala mutant, with quantification using the Bratton-Marshall assay. As shown in (**g**), 4-nitrobenzoic acid is reduced to 4-aminobenzoic acid, and 2-nitroimidazole is reduced to 2-aminoimidazole in (**h**). **i** LC-MS/MS quantification of 2-aminoimidazole formed from the reduction of 2-nitroimidazole in nitroreductase assays with wildtype or His55Ala *Cd*NimBs. There are differences in relative amounts of 2-aminoimidazole quantified by the LC-MS/MS and Bratton-Marshall methods, but the results from both reached the same conclusion that the mutant is less effective in forming the amine product. **j** Cellular reduction of 2-nitroimidazole to 2-aminoimidazole. Concentrated cultures of various isogenic strains were treated with 2-nitroimidazole (2 mM) alone or with heme and incubated for 3 h, before 2-aminoimidazole were quantified. Plots show the mean ± standard error of mean from four replicates, except for three biological replicates for R20291Δ*nimB* comp.P*nimB*. Statistical analyses in (**g**)–(**j**) were done by two-tailed unpaired *t* test in Graphpad prism 9.4.1. Data in (**c**), (**f**)–(**i**) are from three biological replicates. Data in (**d**) and (**e**) show one of three biological replicates tested; the other two replicates behaved as shown in (**d**) and (**e**).

To quantify amino reaction end-products of *Cd*NimB enzymatic and cellular nitroreductase activities we adopted the related 2-nitroimidazole, since 5-aminoimidazoles derived from 5-nitroimidazoles (e.g., metronidazole) are chemically unstable[51,52], which would present challenges to quantify by LC-MS/MS. In contrast, 2-nitroimidazole produces more stable 2-aminoimidazoles[52] and is also a substrate for *Cd*NimB (Fig. 3f); nitrobenzoic acid lacked antimicrobial activity against *C. difficile* cells. *C. difficile* also showed heme-dependent resistance to 2-nitroimidazole i.e., in the absence of heme, MICs were 0.25–1, 0.25–0.5, and 0.25–1 μg/ml, against susceptible CD196, R20291Δ*nimB* and R20291-Tn::*nimB* (Supplementary Fig. 6b). Complementation with wildtype *nimB* from R20291 conferred resistance to 2-nitroimidazole in the presence of heme (MICs of 4–8 μg/ml), but complementation with the Ala-55 variant did not restore resistance to wildtype levels (MICs of 0.25–2 μg/ml; Supplementary Fig. 6b). In nitroreductase enzyme assays, NimB-Ala-55 was less efficient in reducing 4-nitrobenzoic acid and 2-nitroimidazole (using the Bratton-Marshall assay, Fig. 3g, h). LC–MS/MS analysis confirmed that 2-nitroimidazole was converted to 2-aminoimidazole by wildtype *Cd*NimB, whereas the NimB His55Ala variant made considerably less of the amino product (Fig. 3i). We next examined cellular reduction of 2-nitroimidazole, using isogenic strains of R20291 (with an empty vector control, R20291Δ*nimB*, and R20291Δ*nimB::nimB* complementation) and CD196 (with an empty vector control or ectopically expressing *nimB* from P*nimB^G*). In the absence of heme, both susceptible and resistant strains produced close to equal, but low, amounts of 2-aminoimidazole (~419 ± 48 to 542 ± 24 μM), as measured by LC-MS/MS. Adding heme increased the production of 2-aminoimidazole by ~60.9–77.8% (i.e., ~1386 ± 201 to 1890 ± 340 μM), but in only the resistant strains (Fig. 3j). Furthermore, when isogenic strains of R20291-Tn::*nimB* were treated with 2-nitroimidazole and heme, the strain complemented with wildtype *nimB* showed 25.4–29.0% more aminoimidazole production than strains complemented with the Ala-55 codon variation or the empty vector (Supplementary Fig. 9). Taken together, these findings indicate that *Cd*NimB is a nitroreductase that converts nitroimidazoles to their amino derivatives both enzymatically and cellularly, in a heme-dependent manner.

## Heme-dependent metronidazole resistance is displayed by *B. fragilis nimA*

Since the above observations might contribute to resolving a long-standing debate on whether *nim* genes confer metronidazole resistance[53], we tested whether *nimA* from *B. fragilis* also confers heme-dependent resistance. As such, codon-optimized *B. fragilis nimA* (OCL16839.1) engendered heme-dependent resistance when expressed in metronidazole-susceptible *C. difficile* CD630 and R20291Δ*nimB* or it elevated resistance in R20291 (Supplementary Fig. 10a). Similarly, assessment of the *nimA* reference strain *B. fragilis* 638 R/pIP417(*nimA*)[54] showed it displayed heme-dependent resistance to metronidazole (MICs of 0.5–1 μg/ml and 4–8 μg/ml, without and with heme, respectively; Supplementary Fig. 10b). We therefore speculate that, like *nimB* from *C. difficile*, *nim* genes in other organisms may be cryptic and their resistance phenotypes may be dictated by cellular regulation of *nim*, and the intracellular availability of cofactors (i.e., heme and possibly flavin).

## Genetics of metronidazole resistance in natural populations

Since heme-dependent metronidazole resistance appears to be the dominant phenotype of metronidazole-resistant *C. difficile*, we decided to further investigate potential mechanisms underlying this phenotype with genome-wide association studies (GWAS) of geographically diverse natural bacterial populations. We performed these analyses on a dataset of 348 clinical isolates for which we determined MICs of metronidazole in the presence and absence of heme. We used two methods to identify genetic variants associated

with heme-dependent metronidazole resistance: FST outlier analysis[55], which identifies alleles with marked variation in frequency among sub-populations, and PySeer, which uses linear models with fixed or mixed effects to estimate the consequence of genetic variation in a bacterial population on a phenotype of interest[56].

Both methods identified two SNPs as extreme outliers in their association with metronidazole resistance (Fig. 4a, Supplementary Fig. 11). The *nimB*-promoter mutation was one of these SNPs and, surprisingly, the other variant associated with resistance was a mutation 245C>T in *gyrA* (Thr82Ile in gyrase A) that confers resistance to fluoroquinolones. The Thr82Ile mutation has been associated with pandemic spread of health care-associated *C. difficile*[24]. There is no known mechanistic basis for an association between *gyrA* and metronidazole resistance. Instead, we hypothesize that the Thr82Ile in GyrA and the *nimB*-promoter SNP are both under positive directional selection, and that their combination could have enabled pandemic spread of *C. difficile*. To test this hypothesis, we first investigated the genomic data for signs of selection by identifying homoplasies: variants that arise independently more than once on the phylogeny and are a potential signal of positive selection[55]. Using ancestral reconstruction, we estimated that the *gyrA* mutation arose independently on 17 occasions in our sample of 348 strains, whereas the *nimB*-promoter mutation arose 15 times, placing both variants in the 99.9th percentile of homoplasies. These findings are indicative of positive selection and suggest that the use of metronidazole, in addition to fluoroquinolones, exerted important selection pressures shaping recent adaptation of *C. difficile*.

A maximum likelihood phylogeny inferred from whole genome sequence data is shown with associated metronidazole susceptibility, Clade and *gyrA*/*nimB*-promoter allele indicated for each isolate in the sample (Fig. 4b). The phylogeny shows that the P*nimB^G* mutation occurs almost exclusively in association with the *gyrA* mutation. While more isolates (*n* = 34) carried the *gyrA* mutation alone than the *nimB* mutation (*n* = 2), most isolates with the *gyrA* mutation also carried the P*nimB^G* variant. *C. difficile* can be clustered into five major Clades[5]. In our sample, these co-occurring variants were primarily associated with Clade 2 and to a lesser extent Clade 1, whereas they were absent from Clades 3 and 5 (Fig. 5g).

Within Clades 1 and 2, the *gyrA* and P*nimB^G* mutations were associated with large clonal groups (Fig. 4b), suggesting that strains with these variants were rapidly transmitted. To test this hypothesis, we re-analyzed the global dataset (n = 119) from ref. 24 who previously determined that *gyrA* mutations were associated with global epidemic spread of health care associated *C. difficile*. Our re-analysis of these published genomes showed that all but one of the isolates carrying the *gyrA* mutation also carried the P*nimB^G* variant. Together, these observations suggest that mutations at the *nimB*-promoter site are responsible for the vast majority of metronidazole resistance (Fig. 5b, c) and thus likely played a pivotal role in declining efficacy of this treatment. We further link metronidazole resistance with fluroquinolone resistance and find evidence suggesting the combination of resistances underlies the recent global spread of health care associated *C. difficile* strains, particularly those belonging Clade 2.

Our study also identified heme-dependent metronidazole-resistant strains that lacked the P*nimB^G* SNP (Supplementary Data 1). Molecular analysis of two of these strains (i.e., 17/27 [RT001] and 25603 [RT137]) showed that their resistance phenotype is also influenced by *nimB*. In both strains, *nimB* mRNA levels were higher than those in susceptible CD196 or CD630 (Supplementary Fig. 12a, b). When *nimB* was silenced using pMSPT (the paired-termini ATc inducible system)[42], both strains became more susceptible to metronidazole (Supplementary Fig. 12c). The MIC of 25603, was reduced from 2 to 4 μg/mL to 0.25–0.5 μg/mL, similar to RT027 strains tested in Fig. 2. However, MICs against 17/27 were reduced from 8–16 μg/mL to 2–8 μg/mL (median of 4 μg/mL, Supplementary Fig. 12c). It is possible that 17/27

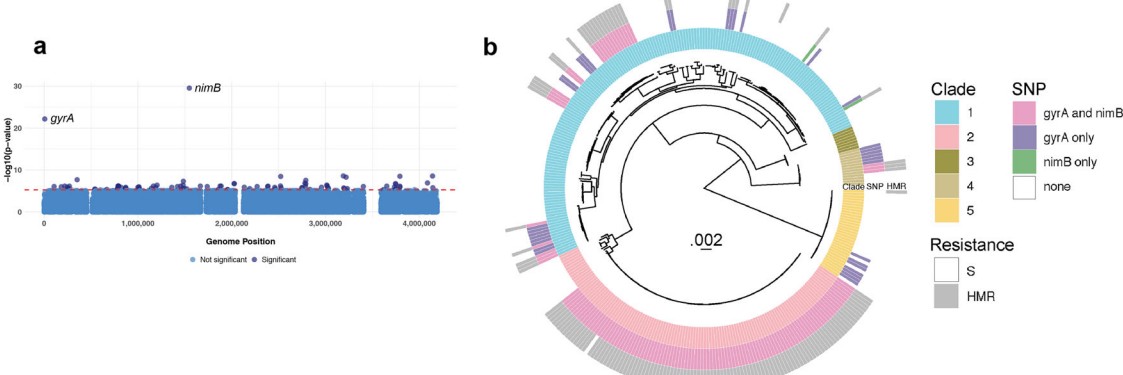

**Fig. 4 | P*nimB^G*-associated metronidazole (MTZ) resistance co-occurs with flruoquinolone resistance and pandemic spread. a** Manhattan plot showing SNPs significantly associated with heme-induced metronidazole resistance (HMR). Adjusted, log-transformed *p* values calculated by pySEER are plotted by genome position; default statistical analyses built in pySEER were used. Both *gyrA* and *nimB* SNPs are very significantly associated with HMR. **b** Maximum likelihood phylogeny based on whole genome sequences from 348 isolates. Multi-locus sequence type

(MLST) Clades 1-5 (inner circle), *gyrA* and *nimB*-promoter mutations (middle circle), and metronidazole susceptibility (outer circle), either heme-induced metronidazole resistant (HMR) or metronidazole susceptible (S), are shown. The *gyrA*, *nimB*-promoter mutations and associated HMR are associated with clonal groups in Clades 2 and 1, suggesting that bacteria carrying these variants have spread rapidly and/ or are sampled more densely in this population.

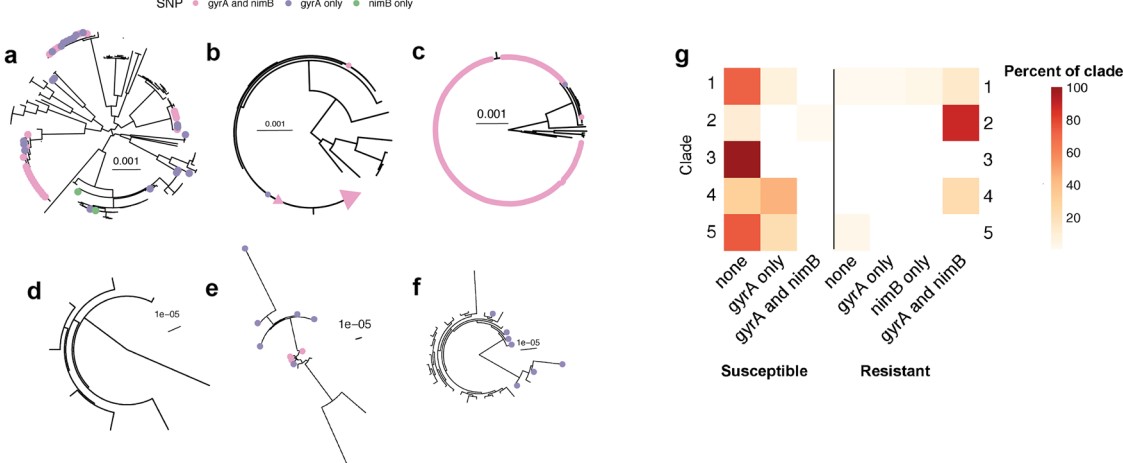

**Fig. 5 | Individual clade phylogenies.** Maximum-likelihood phylogenies of individual Clades 1–5 with tips colored according to *gyrA* and *nimB* SNP presence or co-occurrence from this study as well as ref. 24. **a** Clade 1; **b** Clade 2, including "global" isolates from ref. 24 and those in this study. The pink triangle represents the clonal group of which all but one have both SNPs; (**c**) Inset showing detail of the clonal group within Clade 2 (with one exception (blue dot; *gyrA* only), isolates with the *gyrA* mutation also encoded the *nimB* mutation; **d** Clade 3; **e** Clade 4; and **f** Clade 5.

**g** Heat map showing SNP presence/co-occurrence and MTZ resistance phenotype frequencies within each Clade of only our sample. Each row corresponds to a single Clade. Each column presents a summary of *gyrA* and *nimB* SNP presence and MTZ resistance phenotype. Heat map is shaded by the percentage of each condition within each Clade, such that the sum across a single row is 100%. 88% of Clade 2 isolates are HMR with both *gyrA* and *nimB* SNPs, while 100% of Clade 3 isolates are MTZ susceptible and have neither SNP of interest.

has additional metronidazole resistance mechanisms. Other chromosomally mediated mechanisms of metronidazole resistance in *C. difficile* include mutations to PFOR, manipulations of iron and electron transport metabolisms[18,57]. Alternatively, it may be possible that the antisense in vector pMSPT is not optimal across different ribotypes. Nonetheless, these results show that while most heme-dependent metronidazole-resistant strains encode P*nimB^G*, a minority of these strains appear to have alternative transcriptional or posttranscriptional mechanisms that regulate *nimB*.

## Discussion
Over the last two decades, the global spread of epidemic *C. difficile* has been paralleled by declining efficacy of metronidazole, culminating in major guidelines removing it as a recommended first-line option for CDI. We recently demonstrated that heme-dependent resistance to metronidazole, defined by an MIC breakpoint of ≥1 µg/mL, was

associated with an increased risk of metronidazole clinical failure[58]. Fundamental to this finding was the revelation that heme is crucial for detecting metronidazole-resistant *C. difficile*[22,23]. Our findings that most metronidazole-resistant strains evolved a T to G mutation (P*nimB^G*) in the −10 promoter of *C. difficile nimB* provide a microbial genetic explanation as to why such strains might be less responsive to metronidazole therapy. Additionally, the fact that P*nimB^G* co-occurs with the Thr82Ile substitution in GyrA updates the paradigm for the global dissemination of epidemic strains[24], suggesting that CDI outbreaks were mediated by strains with resistance to not only fluoroquinolones, but also to metronidazole. These dual drug resistances likely afforded epidemic *C. difficile* an advantage in healthcare settings. Multiple lines of evidence support P*nimB^G* as a cause of widespread metronidazole resistance, including our GWAS using two independent methodologies (Fig. 4a, Supplementary Fig. 11, a previously published GWAS of independent samples[59], our documentation of the mutation

emerging in vivo in a metronidazole-treated patient, and its mechanistic genetic validation as explained below.

Our analyses of population-level genomic data found *nimB* and *gyrA* variants are advantageous, as expected for drug resistance mutations. The two mutations were strongly linked, and phylogenetic analyses suggest that strains carrying both mutations spread more rapidly in healthcare settings (Figs. 4b, 5a–f). This hypothesis is consistent with previously published data suggesting that fluroquinolone-resistant *C. difficile* spread rapidly across continents[24–26]. We found the combination of *nimB* and *gyrA* mutations in Clade 2 and to a lesser extent Clade 1 (Figs. 4b, 5a–f). This association could reflect niche specialization among the Clades. Large scale surveys of *C. difficile* genomic data have identified high rates of carriage of both antimicrobial resistance determinants and toxin genes in Clade 2[21,59,60]. This genetic cargo reflects adaptation of Clade 2 to the antibiotic milieu of health care settings and pathogenicity in human diarrheal disease[21,25]. Clade 5 is similarly associated with antibiotic resistance and increased virulence[59], but we did not identify the *nimB*-promoter mutation in our sample of Clade 5 isolates (Fig. 5f). Clade 5 antimicrobial resistance determinants reflect the livestock and farming environments that are associated with this Clade[21] and absence of the *nimB*-promoter mutation from this Clade could reflect the distinct antibiotic selection pressures encountered in these environments. It is also possible that the *nimB*-promoter mutation, and resulting heme-dependent metronidazole resistance, is a human-specific adaptation. Given the small number of Clade 5 isolates included in our analyses, a larger sample size will be required to test this hypothesis. An alternative, but not mutually exclusive, explanation for the observed associations between the *nimB*-promoter mutation, the *gyrA* mutation and the Clade 2 genetic background is epistasis, i.e., interactions among genetic loci that affect cell physiology. It is striking that the *nimB* mutation almost exclusively appears among strains with the *gyrA* mutation (Fig. 4b). A potential explanation for this phenomenon is that the *nimB*-promoter mutation and constitutive formation of *nimB* imposes a metabolic burden and fitness costs that might be ameliorated in strains with the *gyrA* mutation. However, isogenic 23468 and 23475, which are resistant and susceptible respectively, did not show significant differences in growth rates (Supplementary Fig. 3). On the other hand, Thr82Ile mutation in GyrA either marginally enhances fitness or has no fitness costs[61,62]. Future studies will be required to determine the extent to which coexisting metronidazole and fluoroquinolone resistance affects *C. difficile* transmission, using clinically reflective animal and in vitro models.

We discovered that *Cd*NimB acts as a nitroreductase both enzymatically and in cells and requires heme to effectively produce amino end products. According to the accepted model, Nim proteins convert nitroimidazoles to amine by rapidly transferring a total of six electrons from unresolved cofactors to the nitro group[35,45]. This requirement for heme may explain why Nim proteins have not been reported to exhibit nitroreductase activity in enzymatic assays but was seen cellularly in heme-containing broths[35]. Nim proteins are structurally and functionally homologous to the flavoenzyme Anf3, in which neighboring heme and flavin binding domains are thought to facilitate rapid electron transfer between the two cofactors. Based on the homodimeric homology model from Anf3 structure (Fig. 3b), and biochemistry of heme flavoenzymes, we speculate *Cd*NimB may reduce the nitroimidazole nitro-groups with electrons from heme or flavin cofactors in the homodimeric protein. Further research is required to test these hypotheses in more optimal biochemical and biophysical experiments to establish the molecular mechanism(s) through which Nim proteins reduce nitroimidazoles. As our experiments used millimolar substrate concentrations, these further studies should include an analysis of the enzyme's kinetic parameters. This will provide a more accurate assessment of the enzyme's catalytic efficiency in reducing physiologic, micromolar concentrations of metronidazole. Overall, our study

identified for the first time *nimB* as a cryptic metronidazole resistance gene linked to the *C. difficile* epidemic. Noteworthy, intravenous metronidazole is still recommended with vancomycin for fulminant CDI by IDSA/SHEA and ESCMID[7,8] and metronidazole remains commonly used in the United States[63] and in other countries where it is still recommended to treat CDI[64,65]. Therefore, this study provides several key insights that could be applied to modernize the use of metronidazole, as well as other CDI medications, through evidence-based approaches. To achieve this, rapid susceptibility testing, genomic surveillance, and mechanistic research will be required to delineate how resistance alleles influence clinical outcomes.

## Methods

### Bacterial strains
*C. difficile* strains used can be found in Supplementary Data 1, including accession numbers of genomes deposited in NCBI database. The geographic locations from which strains were isolated are specified: for GWAS strains were isolated in United States, Europe and Israel and other locations, as documented; other strains were from Texas Medical Center. Also included were strains reported from the United States Centers for Disease Control and Prevention Emerging Infections Program[66]. *B. fragilis* 638R/pIP417(strain DSM103646) and DSM2151 were from Leibniz Institute DSMZ; DSM2151 is an antibiotic susceptibility testing control strain. All strains of *C. difficile* or *B. fragilis* were cultured in pre-reduced Brain Heart Infusion (BHI) broth overnight at 37 °C in a Whitley A35 anaerobic workstation (Don Whitley Scientific).

### Determination of minimum inhibitory concentrations (MICs) by agar dilution method
MICs were determined by the agar dilution method on BHI agar either with or without 5 μg/ml of porcine hemin (Alfa Aesar, catalog no. A11165) and doubling dilutions of metronidazole (0.125–32 μg/ml; from Acros, catalog no. 210340050), as previously described[23] by inoculating agars with 2 μl of overnight cultures that contain $10^4$–$10^5$ CFU/spot. Metronidazole susceptibility tests in BHI were shown to be comparable to Wilkins Chalgren and Brucella agars[23]. Metronidazole and hemin stocks were prepared in dimethyl sulfoxide (DMSO) (Alfa Aesar, catalog no. 43998). Plates were incubated for up to 48 h and MICs recorded as the lowest concentration of drug that inhibited visible growth. MICs for 2-nitroimidazole (Sigma-Aldrich, catalog no.195650) were similarly performed.

### Generation of transposon mutagenesis library of metronidazole resistant *C. difficile*
Plasmid pRPF215, a *Himar1* mariner delivery vector[30], was conjugated into R20291 from *E. coli* SD46 and R20291 transconjugants selected on BHI agars supplemented with 250 μg/ml cycloserine (Matrix Scientific, catalog no. 072929), 8 μg/ml of cefoxitin (Chem-Impex International, catalog no. 01490), and 15 μg/ml thiamphenicol (Sigma-Aldrich, catalog no. T0261). To generate Tn mutants, R20291-pRPF215 was grown overnight in BHI broth with 15 μg/ml thiamphenicol, then subcultured into BHI to $OD_{600}$nm of 0.2. *Himar1* was induced with 100 ng/ml anhydrotetracycline (ATc) (Alfa Aesar, catalog no. J66688) for mutagenesis for overnight (15 h). Tn mutants were selected by spreading culture dilutions onto pre-reduced BHI agar containing 20 μg/ml lincomycin. After incubation (24 h), colonies were collected into 96 deep well plates, to amass 7488 independent colonies. The library was screened for metronidazole-susceptible Tn mutants by growing overnight cultures in BHI containing lincomycin and spotting 2 μl inocula onto BHI agars containing 0.5 μg/ml metronidazole and 5 μg/ml of hemin using a 96-Well Bench Top Pipettor. Presumptive metronidazole-susceptible Tn mutants were confirmed by MIC testing and underwent genome and Sanger sequencing to confirm the insertion sites for *ermB*. Primers and strain constructs used are listed in the Supplementary Data 2 and 3.

## Generation of knock out strain

The vector pMTL-SC7215-CD2517.1 was created by modifying vector pMTL-SC7215 replacing the cytosine deaminase (*codA*) gene with *CD2517.1* Type I toxin-antitoxin system as counter-selectable, essentially as previously described[67]. To delete *nimB* by allelic exchange, a 1000 bp cassette of the upstream and downstream flanking regions were synthesized by Genscript, cloned into pMTL-SC7215-CD2517.1 and conjugated into R20291. After eight serial passages in BHI, cultures were plated onto agars with ATc (70 ng/ml) and colonies were screened by PCR and Sanger sequencing for loss of *nimB*.

## Overexpression and knockdown of target genes

To overexpress *nimB* under its indigenous promoter, *nimB* together with its 504-bp upstream sequence was cloned between KpnI and BamHI sites of vector pRPF185, producing pRPF185-P*nimB*. Overexpression of *nimB* under anhydrotetracycline-inducible promoter (Ptet) was done by cloning the gene with its ribosome binding site into pRPF185, producing pRPF185-P$_{ATc}$*nimB*. Plasmid constructs were conjugated into *C. difficile* from *E. coli* SD46 and strains were grown in BHI broth with 15 μg/ml thiamphenicol. MICs were performed as previously indicated but in plates supplemented with thiamphenicol (15 μg/ml) and ATc (32 ng/ml) specifically for pRPF185-P$_{ATc}$*nimB*. Translational knockdown was performed as described[42], whereby a 100-bp antisense RNA synthesized by Genscript, spanning 50-bp upstream and downstream of *nimB* from the start codon, was cloned into SphI and XhoI sites of vector pMSPT. Induction of gene knockdown was performed with 64 ng/ml of ATc. CRISPR interference was done as described[42], whereby the guide RNA targeting *nimB* was synthesized by Genscript and cloned into the PmeI site of vector pXWxyldcas9. The guide RNA was induced with 1% xylose (Alfa Aesar, catalog no. A10643). Primers and strain constructs used are listed in the Supplementary Data 2 and 3.

## Promoter strength analysis

The 504-bp upstream region of *nimB* was cloned into NheI and SacI sites of pDSW1728 vector to generate P*nimB*$^G$::mCherryOpt or P*nimB*$^T$::mCherryOpt from R20291 or CD196 respectively. Constructs and the empty vector control were conjugated into various *C. difficile* strains. Strains bearing the reporter construct and empty vector were grown to OD$_{600}$ nm ~ 0.3, from overnight cultures. Each culture was either treated with DMSO control, metronidazole (2 μg/ml), metronidazole (2 μg/ml) plus hemin (5 μg/ml) or hemin alone (5 μg/ml) and incubated anaerobically for 1 h. Samples were then fixed as described previously[41] and fluorescence measured in BioTek Synergy reader at excitation of 554 nm and emission of 610 nm. The fluorescence value for each sample was then normalized based on their culture densities (OD$_{600}$ nm values). Primers and strain constructs used are listed in Supplementary Data 2 and 3.

## Construction and synthesis of Nim-like homologs

Genes encoding Anf3 and NimA from *A. vinelandii* and *B. fragilis*, respectively, were codon optimized for translation in *C. difficile* using JCat platform[68]. A synthetic ribosome-binding site for each gene was designed using RBS Calculator v2.1[69] and placed ahead of the start codon and the ensemble synthesized by Genscript. They were then subcloned into pRPF185 vector and expressed from the *Ptet* promoter using 8 ng/ml of ATc. Primers and strain constructs used are listed in Supplementary Data 2 and 3.

## RNA sequencing

R20291 was grown to OD$_{600}$ nm of 0.2 in BHI broth and exposed to DMSO control, metronidazole (2 μg/ml), metronidazole (2 μg/ml) plus hemin (5 μg/ml), or hemin alone (5 μg/ml) for 30 min. RNA was extracted using Qiagen RNeasy miniKit (Qiagen, Valencia, CA) and treated with TURBO™ DNase (Thermofisher Scientific). RNA

sequencing was then performed by GENEWIZ, LLC. (South Plainfield, NJ, USA), by 2 × 150 Paired End (PE) configuration by Illumina HiSeq. Raw RNA-seq data were uploaded onto galaxy platform (https://usegalaxy.org/). Quality control was carried out using FastQC (galaxy version 0.72+galaxy1) and multiFastQC (galaxy version 1.7). Trimming was done using Trim Galore (galaxy version 0.4.3.1) to remove adapter sequences. Reads were mapped to R20291 reference FN545816.1 using the BWA-MEM program (galaxy version 0.7.17.1). Counts per Read was generated using htseq-count (galaxy version 0.6.1galaxy3), while differential gene expression was done with Degust (http://degust.erc.monash.edu/) using edgeR (cut-offs absolute log$_2$ ≥ 1 and FDR ≤ 0.01).

## Reverse transcription quantitative real-time PCR

From total RNA, cDNA synthesis was done using qScript cDNA Supermix kit (Quantabio). qPCRs were carried out in 20 μL reactions, using qScript 1-Step SYBR Green qPCR kit (Quantabio) on a ViiA 7 Real-Time PCR System (Applied Biosystems); 16S rRNA was used for normalization. Gene expression was calculated by the standard ΔΔCT method. Constitutive gene expression analysis was calculated relative to metronidazole-susceptible strains. Primers used are listed in the Supplementary Data 2.

## Structural modeling of NimB

A homology model of R20291 *Cd*NimB was generated from *B. thetaiotaomicron* NimB dimeric protein structure with bound FAD (PDB 2FG9, resolution 2.20 angstrom [https://www.rcsb.org/structure/2FG9]). Prime software in the 2020-2 Schrödinger molecular modeling suite was used to build and refine the model[70,71]. The *Bt*NimB and *Cd*NimB models were aligned to Anf3 nitrogenase from *A. vinelandii* (PDB 6RK0, resolution 0.99 angstrom[43] [https://www.rcsb.org/structure/6RK0]) using the Schrödinger software.

## Site directed mutagenesis of NimB

Heme-binding residues were predicted through alignment of *Cd*NimB with known heme-binding pyridoxamine 5′-phosphate oxidase family proteins, analysis in hemoquest and structural comparison with Anf3. Point mutations were generated in pRPF185-P*nimB* by standard DpnI mediated site-directed mutagenesis, variants were confirmed by Sanger sequencing and conjugated into *C. difficile*. Susceptibilities to metronidazole or 2-nitroimidazole were determined as described above. Primers used are listed in Supplementary Data 2.

## Purification of NimB proteins

Genes encoding NimB or NimB-His55Ala variant were cloned into pWL613a, a derivative of pET28b plasmid. They were transformed into *E. coli* BL21 (DE3), grown in Terrific Broth with 8 μM hemin and induced with 0.4 mM IPTG at 14 °C for 18 h. Harvested cultures were suspended in buffer (50 mM Tris-HCl (pH8.0), 0.5 M NaCl, 10 mM imidazole, 18% glycerol, 5 mM *β*-mercaptoethanol, 1X protease inhibitor cocktail, 0.01 mg/ml DNase, 8 μM hemin and 5 mM MgCl$_2$), lysed by French press 28,000 psi, His-tagged proteins were purified from Ni-NTA columns (Marvelgent Biosciences Inc), and dialyzed for 20 h at 4 °C in buffer [20 mM Tris-HCl (pH8.0), 0.25 M NaCl, 18% glycerol and 5 mM *β*-mercaptoethanol].

## Analysis of heme binding

Nim proteins were purified as above, but without hemin being added to the *E. coli* culture and lysis buffer. To determine if the proteins were purified with bound heme, samples were tested using the Hemin Assay Kit (Sigma MAK036) per the manufacturer's protocol. Dialyzed samples of wildtype and Ala-55 *Cd*NimB 50 μl were mixed with 50 μl of the reaction mixture in 96-well microtiter plates. The reaction was incubated in the dark at room temperature for 30 min, before absorbance was read at 570 nm. The concentration of bound hemin/heme was determined from a standard curve of hemin; DNase served as a

negative control. Results provide a semi-quantitative measure of heme that becomes available from the test enzymes to peroxidase in the Hemin Assay Kit.

The ability of Nim to bind heme was determined by absorption spectroscopy[72] in which the protein is titrated with heme. To determine heme binding, protein (10 μM) in buffer (20 mM Tris-HCl [pH 8.0], 0.25 M NaCl, 18% w/v glycerol and 5 mM $\beta$-mercaptoethanol) was used to collect the absorbance spectra between 350 and 700 nm at 2 nm increments in polystyrene 24-well plates in the dark. The protein was titrated by stepwise addition of hemin from 1 mM stock in DMSO and absorbance spectra recorded after 5 min of incubation at room temperature using Synergy H1 BioTek microplate reader. Titration was continued until there was no further significant change in the spectra, indicating saturation. The spectra of the buffer containing hemin was recorded under the same conditions and subtracted from the protein samples. Heme binding was analyzed in plots of the change in absorbance at 416 nm (Soret band) against the hemin concentration, and qualitatively by visualizing pooled samples.

### Nitroreductase enzymatic assay

NimB nitroreductase activity was assayed using substrates 4-nitrobenzoic acid, 2-nitroimidazole, metronidazole, dimetridazole with controls as sodium benzoate, imidazole, and vancomycin. 4-nitrobenzoic acid and imidazole were from Alfa Aesar, catalog nos. A14738 and A10221, respectively; vancomycin, dimetridazole and sodium benzoate were from Sigma-Aldrich catalog nos. V2002, D4025 and 109169, respectively. The assay was done in an anaerobic chamber. The reaction was performed in 96-well microtiter plates containing 50 mM KPO$_4$ buffer with 10 μM hemin, 3 mM NADPH (Cayman, catalog no. 9000743), 10 μM FAD (Cayman, catalog no. 23386), 7.5 U/ml pyranose oxidase (Sigma-Aldrich, catalog no. P4234), 1 KU/ml catalase (Sigma-Aldrich, catalog no. C9322), 50 mM glucose and 5 mM substrate; workup experiments tested NADPH at 0.3–6 mM. Pyranose oxidase catalase was used to remove residual oxygen, as previously reported[43], whereby the assembled components (except CdNimB) were pre-incubated at 37 °C for 1 h. Next, addition of CdNimB, wildtype or His55Ala variant, was done to a final concentration of 9.7 μM and incubation continued for 2 h. The reaction used 5 mM of substrate to ensure detectable levels of 5-aminoimidazole product from metronidazole, since micromolar concentrations of metronidazole did not result in amine detection, likely due to the known chemical instability of 5-aminoimidazole[51,52]. This concentration was also used as a standard concentration for all other test substrates and controls. Primary nitroaromatic amines were detected using the Bratton-Marshall assay reagents[73,74]. Detection and quantification of 2-aminoimidazole formation from 2-nitroimidazole reduction was also done by LC-MS/MS, as 2-aminoimidazole is considerably more stable than 5-aminoimidazoles.

**Bratton–Marshall detection of nitroaromatic amines.** In the anaerobic chamber, an equal volume of 20% cold Trichloroacetic acid (TCA) (Sigma-Aldrich, catalog no. T0699) was added to the reaction mixture, and after 30 mins on ice, 200 μL of supernatant was mixed with 25 μL of 0.1% sodium nitrite (Alfa Aesar, catalog no. A18668). After 10 min, 25 μL of 0.5% w/v ammonium sulfamate (Alfa Aesar, catalog no. A17696) was added followed by 25 μL of 0.05% N-(1-Naphthyl)ethylenediamine dihydrochloride (Alfa Aesar, catalog no. A17164). After 20 min, absorbance was measured at 540 nm with a VERSA Max microplate reader. Standard curves were generated using 4-aminobenzoic acid (Sigma-Aldrich, catalog no. A9878) and 2-aminoimidazole (Sigma-Aldrich, catalog no. CDS020502) prepared in DMSO; no standard curve could be generated for 5-aminoimidazole as this chemical is unstable and commercially unavailable.

**LC-MS/MS quantification of 2-aminoimidazole from enzymatic and cellular assays.** After incubation of the enzymatic or enzyme free

reactions, samples were precipitated with cold TCA (10 % v/v). After 30 min on ice, supernatants were collected, stored at −80 °C and LC-MS/MS performed at Baylor College of Medicine (BCM) NMR and Drug Metabolism Advanced Technology Core or Texas A&M University (TAMU) Chemistry Mass Spectrometry Facility. This method was also used to measure formation of 2-aminoimidazole in cells as follows. Logarithmically growing cells (OD$_{600}$nm ~0.3) were concentrated 20-fold and exposed to DMSO control, 2-nitroimidazole (2 mM), 2-nitroimidazole (2 mM) plus hemin (5 μg/ml) or hemin alone (5 μg/ml) and incubated for 3 h anaerobically. After incubation, cells were treated with TCA (10 % (v/v) and supernatants recovered. At BCM, LC-MS/MS was done by diluting samples in methanol. 2-aminoimidazole was resolved, identified, and quantified by UHPLC coupled with Q Exactive Orbitrap MS (Thermo Fisher Scientific) equipped with 50 mm × 4.6 mm column (XDB C-18, Agilent Technologies). MS data were acquired from 50 to 1000 Da in profile mode and reference ions at $m/z$ 371.1012. A standard curve was generated for 2-aminoimidazole to determine concentrations in samples and results were normalized per protein content. Results from LC-MS/MS at BCM are shown in Fig. 3j. At TAMU, LC-MS/MS was done by diluting and filtering samples in acetonitrile. Samples were analyzed using a Thermo Scientific Q Exactive Focus coupled with LC unit (ultimate 3000 RS), equipped with an Accucore-150-Amide HILIC (2.1 × 150 mm; 2.6 μm) column (Thermo Scientific). The Q Exactive Focus ESI source was operated in full MS (50–200 m/z), with mass resolution tuned to 70000 FWHM at m/z 200. Quantitation of 2-aminoimidazole was performed by plotting peak area of the extracted ion chromatogram ($m/z$ 84.0556) against the concentration of the corresponding calibration standard. Results from LC-MS/MS at TAMU are shown in Fig. 3i and Supplementary Fig. 9.

### Whole-genome sequencing

For genome-wide association studies and population genetics, whole genome sequencing was conducted on 365 C. difficile clinical isolates. Of these isolates, 99 were sequenced at University of Texas Dallas Genome Center from multiplexed DNA libraries prepared by the Illumina DNA Prep Kit and sequenced by paired-end 2 × 150 bp on an Illumina Next-Seq™500 platform. The remaining 266 isolates were sequenced at SeqCenter, LLC Pittsburgh, using an Illumina DNA Prep kit and IDT 10 bp UDI indices and sequenced on an Illumina NextSeq 2000 producing 2x151bp reads. Post-sequencing, the metrics from the sequencers were used to guarantee that the number of bases with a quality score of Q30 or higher meets or exceeds the total ordered. The data was demultiplexed and adapters removed using bcl2fastq (v2.20.0.422). Other isolates sequenced in this study were done as above at SeqCenter, LLC.

### Reference guided assembly

We conducted reference-guided assembly on 365 U.S. C. difficile isolates and 136 global, Clade 2, 027/BI/NAP1 C. difficile isolates from ref. 24. We used reference-guided assemblies instead of de novo assemblies to better query variations in intergenic regions. Raw sequencing data were assembled using a reference-guided assembly pipeline (https://github.com/pepperell-lab/RGAPepPipe.git). Quality of the sequencing data was assessed using FastQC v0.8.11[75], reads were trimmed using TrimGalore v0.6.4 (https://github.com/FelixKrueger/TrimGalore) and then aligned to R20291 reference sequence FN545816.1, using BWA MEM v0.7.12[76]. Alignments were processed using samtools v1.3.1[77], and Picard v1.183 (http://broadinstitute.github.io/picard/) was used to remove duplicates and add read information. Reads were realigned using GATK v3.5[78] and variants were identified using Pilon v1.16[79]. Finally, assembly quality was assessed using Qualimap BamQC[80]. We discarded 18 newly sequenced assemblies and 17 assemblies from He et al. with poor quality assemblies, evaluated by the percent of reads aligned to the reference as <70%. Our final dataset was comprised of 348 C. difficile isolates (347 isolates + 1 reference), plus 119 Clade 2 027/BI/NAP1 isolates from He et al.; average read

depth values of the dataset used in our analyses ranged from 31X to 149X; median coverage ranged from 100X to 300X; and all samples analyzed had >70% alignment to the reference.

## Phylogeny

A phylogenetic tree of the whole-genome sequences of our isolates was inferred using RAxML v8.2.3[81] using the maximum-likelihood method under the General Time Reversible model of nucleotide substitution and the CAT approximation of rate heterogeneity. The tree was visualized in R using ggtree[82]. Clade typing was verified using Clade representatives from ref. 83. A phylogenetic tree made by combining Clade 2 isolates from this study and the whole-genome sequences assembled from ref. 24 was also inferred using RAxML and the GTRCAT model.

## Fst outlier analysis

To identify genetic variants associated with our phenotype of interest, we performed an Fst outlier analysis between susceptible and heme-dependent resistant populations. From the whole-genome alignment of all our isolates we created a VCF using SnpSites v2.4.1[84] and calculated Weir and Cockerham's Fst (wcFst) for bi-allelic SNPs using vcflib (https://github.com/vcflib/vcflib) and bespoke scripts (https://github.com/myoungblom/Evolution_of_the_GGI/tree/main/Figure5). To identify wcFst outliers we repeated our analysis 100X using randomly assigned phenotypes and used the maximum wcFst value observed in this null distribution as a significance cut-off. We also ran pySEER[56] using the fixed effects model to identify SNPs associated with HMR.

## Homoplasy analysis

Homoplasy analysis was conducted as previously described[55] using Treetime[85]. To investigate *gyrA* and *nimB* SNPs for signs of selection, we reconstructed ancestral sequences and identified variants which arose more than once in the phylogeny. Then, we quantified the number of times each variant was observed. The distribution of the number of occurrences for each SNP was evaluated.

## Reporting summary

Further information on research design is available in the Nature Portfolio Reporting Summary linked to this article.

## Data availability

The raw whole-genome sequencing data generated in this study have been deposited in NCBI database under Bioproject numbers PRJNA914992, PRJNA943263, PRJNA555597, PRJNA940988; all accession numbers for individual strains are provided in Supplementary Data 1. Raw RNA-Seq data generated in this study were deposited in NCBI database under accession number PRJNA880780. Source data are provided with this paper. Raw data for histograms in various figures are found in the Source Data. Supplementary Data 2 list primers used in this study. Supplementary Data 3 list genetically constructed strains from this study. pySEER analysis is available at https://github.com/myoungblom/cdiff_gwas. All data and materials in this study are available to any researcher for the purpose of reproducing or extending the study's results. Source data are provided with this paper.

## Code availability

All applicable software and codes used are stated and cited in the Methods, and can be readily accessed through the below links, along with information to operate the tools. The following software were used: GraphPad prism 9.4.1; RBS Calculator v2.1; Phyre2; HemoQuest (http://131.220.139.55/SeqDHBM/); JCat platform (http://www.jcat.de); Prime software in 2020-2 Schrödinger molecular modeling suite; software on Galaxy platform (https://usegalaxy.org/) as follows Trim-Galore (galaxy version 0.4.3.1), FastQC (galaxy version 0.72+galaxy1) and multiFastQC (galaxy version 1.7), htseq-count (galaxy version 0.6.1galaxy3), BWA-MEM program (galaxy version 0.7.17.1); edgeR on

Degust platform (http://degust.erc.monash.edu/); bcl2fastq (v2.20.0.422); Reference-guided assembly pipeline (https://github.com/pepperell-lab/RGAPepPipe.git); Picard v1.183 (http://broadinstitute.github.io/picard/); TrimGalore v0.6.4 (https://github.com/FelixKrueger/TrimGalore); Fst outlier analysis using vcflib (https://github.com/vcflib/vcflib) and bespoke scripts (https://github.com/myoungblom/Evolution_of_the_GGI/tree/main/Figure5); pySEER analysis is available at https://github.com/myoungblom/cdiff_gwas.

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

## Acknowledgements

This work was funded by R01AI139261 to J.G.H. from the National Institute of Allergy and Infectious Diseases at the National Institutes of Health. The funders had no role in study design, data collection, interpretation of the findings, or in the writing and submission of the manuscript.

We are grateful for the receipt of clinical strains from Amos Adler, Tel Aviv University, Israel; Paola Mastrantonio, Istituto Superiore di Sanità, Italy; Dale Gerding, Loyola University Medical Center, United States; Scott Curry, University of Pittsburgh School of Medicine, United States; Mary-Beth Dorr, MODIFY trials program, of Merck & Co., Inc; various participants of the Pan-European Longitudinal Surveillance of Antibiotic Resistance among Prevalent *Clostridium difficile* Ribotypes' Study Group; BEI Resources, National Institute of Allergy and Infectious Diseases at the National Institutes of Health; and United States Centers for Disease Control and Prevention Emerging Infections Program. We are also grateful to Jordan May for technical assistance with genome extractions.

## Author contributions

J.G.H. initially conceived the project. A.O.O., C.D., M.A.Y., M.A.T., C.S.P., and J.G.H. conceived, conducted experimental design, execution, and data analysis, and wrote the original draft. All authors reviewed the manuscript. W.S., A.D., and K.L.P. established genome extraction protocols; W.S. and A.D. contributed to establishing susceptibility testing protocols. K.E.H. conducted molecular modeling of NimB. K.W.G. and A.G.L. provided clinical isolates from Houston and reviewed susceptibility testing results. M.H.W. and J.F. provided clinical isolates from Europe.

## Competing interests

The authors declare no competing interests.

## Additional information

[1]Center for Infectious and Inflammatory Diseases, Institute of Biosciences and Technology, Texas A&M Health Science Center, Houston, TX, USA. [2]Department of Biology, University of Waterloo, Waterloo, ON, Canada. [3]Microbiology Doctoral Training Program, University of Wisconsin-Madison, Madison, WI, USA. [4]Department of Pharmacy Practice and Translational Research, University of Houston College of Pharmacy, Houston, TX, USA. [5]Department of Pharmaceutical Sciences, College of Pharmacy, University of Tennessee Health Science Center, Memphis, TN, USA. [6]Department of Microbiology, Leeds Teaching Hospitals Trust, Leeds, UK. [7]Healthcare Associated Infection Research Group, School of Medicine, University of Leeds, Leeds, UK. [8]Department of Biological Sciences, University of Texas at Dallas, Richardson, TX, USA. [9]Department of Medical Microbiology and Immunology, School of Medicine and Public Health, University of Wisconsin-Madison, Madison, WI, USA. [10]Department of Medicine, Division of Infectious Diseases, University of Wisconsin-Madison, Madison, WI, USA. [11]These authors contributed equally: Abiola O. Olaitan, Chetna Dureja. [12]These authors jointly supervised this work: Caitlin S. Pepperell, Julian G. Hurdle. ✉e-mail: cspepper@medicine.wisc.edu; jhurdle@tamu.edu

