## [Peer Review File · Nature Communications]

Decoding a cryptic mechanism of metronidazole resistance
among globally disseminated fluoroquinolone-resistant
Clostridioides difficileReviewer #1 (Remarks to the Author):

This paper from Olaitan et al. constitutes a remarkable achievement in the understanding of metronidazole resistance in *Clostridioides difficile*. In a previous publication in 2021, the lab of Prof. Hurdle had already demonstrated that metronidazole resistance in the overwhelming majority of resistant *C. difficile* strains is dependent on hemin as a supplement to the growth medium. This paper provides a careful and expansive explanation for this phenomenon. The authors clearly demonstrate that metronidazole resistance in *C. difficile* is mediated by the presumptive nitroreductase NimB. The Nim proteins were originally described to occur in some *Bacteroides fragilis* strains but do also exist in other – non-related bacteria – such as *C. difficile*. Although the *nimB* gene is commonly present in *C. difficile*, the resistant strains shown here have an SNP in the promoter of the gene that enables constitutive expression of *nimB*, resulting in strongly elevated mRNA levels. Further, the authors identified NimB as a heme-binding protein which explains why metronidazole resistance only manifests in the presence of heme. The authors present evidence in favour of the hypothesis that NimB acts as a nitroreductase which detoxifies metronidazole to the harmless cognate aminoimidazole. Finally the authors provide a compelling argument that metronidazole resistance co-evolved with fluroquinolone resistance in isolated clades of *C. difficile*. Overall this is really an excellent study which provides plentiful experimental data to support the notion of NimB as a heme-dependent resistance factor which is selectively overexpressed in strains harbouring the said mutation in the *nimB* promoter region. I do, however, have serious concerns regarding the part dealing with NimB's assumed function as a nitroreductase. My queries are listed below.

Queries block A: Enzymological evaluation of NimB

1. The authors' description of NimB's role as nitroreductase is confusing because they alternately propose the transfer of two and of six electrons to the nitro group of metronidazole. In fact, the reduction of the nitro group to a comparably non-reactive amine requires the transfer of six electrons whereas the transfer of two electrons results in the formation of a toxic nitrosoradical. Indeed, already the original hypothesis as proposed by Leiros et al. 2004 was based on the IMO incorrect assumption that the formation of nitrosoradicals would have a protective effect. The authors show that the nitro groups in the respective nitro drugs (Figure 3) are reduced to their cognate amino groups. Thus, the authors should stick to this explanation dating back to the observations made by Carlier et al. 1997 (paper cited by the authors) and not confuse it with the contrasting hypothesis by Leiros et al. (see Discussion lines 394 and 395).

2. I do not understand how 10 μ M of NimB can be possibly saturated with hemin at a hemin concentration of 4 μ M (Figure 3d), i.e. at a 2.5-fold excess of NimB over hemin. Please explain!

3. To the best of my knowledge the nitroreductase assay as described in this paper and as shown in Figure 3f-h is novel. In light of this its description is fairly scant and lacks mention of necessary controls. Further, some of the details do not add up.

a. Did the authors omit hemin, FAD and NADPH in any control experiments? This would be important for conferring credibility to the model and the assay. If the authors' model of NimB as a hemin-, FAD- and NADPH-dependent nitroreductase really applied, then none of these components could be omitted for aminoaromatic compounds to be formed. This has to be tested.

b. Why was the concentration of nitro compounds so high? 5 mM of metronidazole is a concentration several orders of magnitude higher than the concentrations encountered in vivo (MIC of metronidazole in resistant *C. difficile* 1-16 μ g/ml, i.e. 6 to roughly 100 μ M). Did the authors also test lower concentrations? Importantly, even at this high concentration the amount of amino-metronidazole formed is very small and only slightly above control level (Figure 3f). Since the reaction time was 2h and the concentration of NimB in the reaction mixture amounted to 10 μ M, this indicates a minimal turnover of approximately 1 h⁻¹ when taking metronidazole as substrate. This is close to nil and cannot explain resistance to metronidazole. In this context it is also important to mention that 300 μ M of NADPH as added to the reactions is only sufficient to allow the formation of 100 μ M amino compounds, one NADPH molecule providing two electrons via a hydride transfer. It is unclear to me why the authors applied such a low amount of NADPH in their assay. Finally, please transform the values on the y-axis in Figure 3f into μ M, the mere absorption value is not sufficiently informative!

c. Why was the amount of 2-nitroimidazole formed by the NimB H55A mutant so much lower when

2-aminoimidazole was quantified by Bratton-Marshall assay (Figure 3h) as compared to HPLC (Figure 3i)? In fact, the supposed activity of NimB H55A amounts to 75% of the wildtype enzyme when the reaction product is measured by HPLC (Figure 3h) although this mutation is shown to abrogate metronidazole resistance as conferred by NimB (see Figure 3C). Taken into account this discrepancy, the R20291 Tn::nimB mutant complemented with nimB H55A (Figure 3C) would have been an important addition to the experiment as shown in Figure 3j. According to the data shown in Figure 3c, conversion of 2-nitroimidazole in cells harbouring the nimB H55A mutant should be equal to R20291 Δ nimB. According to the high level of 2- aminoimidazole formation by NimB H55A as determined by HPLC, however, it should be close to the level of wildtype R20291. Further also the MIC for 2-nitroimidazole in the presence of heme was strongly elevated in the metronidazole-susceptible strains CD196 and R20291 Δ nimB when nimB H55A was introduced via plasmid pRPF185 (Extended Data Figure 4).

These issues show that 2-nitroimidazole is a poor substitute for metronidazole which is not surprising given that 2-nitroimidazoles have much higher reduction potentials than 5-nitroimidazoles, including metronidazole. They are more easily reduced than 5-nitroimidazoles and cause different effects. The 5-nitroimidazole dimetridazole as used by Carlier et al. 1997 would have been a much more suitable choice for these experiments.

Queries block B: other queries

1. It would be interesting to see overexpression of nimB plotted against the degree of metronidazole resistance. From the data I could muster in the manuscript no correlation between the degree of resistance and nimB mRNA levels can be inferred. Strain 23468, e.g., displays an almost 20-fold overexpression of nimB in the presence of heme as compared to the parental 23475 strain but is only resistant to a metronidazole conc. of 1 μ g/ml (Figure 2d). Strain 70/76, displaying only an about 5-fold overexpression of nimB as compared to susceptible CD196, is resistant to 8 μ g/ml of metronidazole (see Wu et al., 2021 from the same lab). The same goes for the couple 17/27 and 25603 (Extended Figure 5). Strain 17/27 has much lower expression of nimB than 25603 but is far more resistant. This suggests that NimB levels do not translate into the level of metronidazole resistance as such which complicates the notion of NimB acting directly as a nitroreductase. Higher levels of nitroreductase should lead to higher levels of detoxified metronidazole. This aspect should be carefully discussed by the authors.

2. Contrary to the authors' claim (lines 335-337), the inhibition of nimB expression in strains 17/27 and 26503 (Extended Figure 5d), both having elevated nimB mRNA levels but nor mutation in the promoter region, does not result in the restoration of metronidazole susceptibility to an extent comparable to the strains shown in Figure 2g and h. Surprisingly, antisense mRNA has no effect at all and CrispR only partially restores susceptibility in 17/27. Please explain!

3. The growth experiment with strains 23468 and 23475 as discussed in the Discussion (lines 381-383) is not sufficient to show that nimB overexpression has no fitness costs. The experiment was performed in BHI medium without hemin although hemin is required to observe a phenotype of nimB overexpression. I think the experiment should be presented in the Results section and not in the Discussion.

4. The data shown in Extended Figure 7 and discussed in the Discussion lines 399-403 should also be transferred to the Results. The results with NimA from *B. fragilis* are interesting and mirror our own observations. However, as a word of caution: resistance to metronidazole in Bf 638R pIP417 (nimA) is also clearly increased when the cells are plated on BHI plates with extra FeCl₃ (our own as yet unpublished results) which is not the case in *C. difficile*, at least according to the authors' previous results (Wu et al., 2021). It is therefore uncertain if the mechanism is the same as observed in *C. difficile*.

To sum up, this paper constitutes an important milestone in *C. difficile* research and definitely deserves to be swiftly published. The part on the presumed nitroreductase activity, however, requires much more attention and care before publication. In order to avoid an unduly delay of publication of the manuscript, I would therefore suggest to shorten this part of the manuscript and expand on the issue in a follow-up paper.

Reviewer #2 (Remarks to the Author):

The manuscript "Decoding a cryptic mechanism of metronidazole resistance among globally disseminated fluoroquinolone-resistant *C. difficile*" combines a myriad of methods to dissect a mechanism of resistance to metronidazole that is associated with a *gryA* mutation that was previously shown to facilitate the global dissemination of epidemic Cd strains. The conclusions are in most cases well supported by the data and have multiple theoretical and pragmatical applications.

This reviewer believes that the authors confuse causality with association at some points along the text but this can be addressed by changing the wording.

Here a few suggestions:

Abstract:

indicate %id of *CdnimB* to other *nim* genes from other species.
Include MICs and change folds to appraise the magnitude of the phenotype

Introduction:

l47: belongs to "MLST Clade 2". There are different grouping methods, in this work only MLST is used and this should be indicated for the sake of clarity.

l49. also in Latin America.

l88. Please elaborate the mechanism of action of metronidazole so that the reader can later understand the transcriptomics responses.

l90. "revealing its role in the global transmission" This is speculative, what the authors can claim with the current experimental design is association/linkage, not causality. It would be advisable to present metronidazole consumption figures or usage data to better understand the strength of the selective pressure in the countries where the isolates were recovered.

l99. To ultimately confirm co-selection, additional experiments in, say, a human gut model, are required. This can be discussed.

Results:

l153. please add "in *C. difficile*" before "since".

l159. please add "plasmid" before "plasmid".

l169. for consistency, please indicate ST and MLST clade of isolates 23475 and 23468.

l232 and 234. Homologous refers to common ancestry, here you mean similar 3D structure.

l245. please indicate method.

l339-340. Perhaps knowledge from other bacteria can shed light onto this alternative mechanism, please consider whether this can be discussed.

Discussion:

l352. Indicating? To me the paper can not confirm this claim.

l361. It is a bit reckless to conclude "that strains carrying both mutations spread more rapidly". Our current knowledge on the global population of Cd is highly biased towards a limited number of lineages linked to some phenotypes.

1364, 365, 368, 373. Clade with capital C.

1404. In the abstract the gene is called nimB, here only nim. What's the correct nomenclature and why?

Congratulation on such a great paper, very exhaustive experimental design and a great contribution to the field.

Reviewer #3 (Remarks to the Author):

In this study, Olaitan and colleagues characterise a new mechanism of metronidazole resistance in *Clostridioides difficile*. This is an excellent multi-disciplinary study where authors provide extensive genetic, biochemical and evolutionary evidence of the role of over-expression of NimB in metronidazole resistance. Find below my major and minor comments:

Major comments:

1. Data availability. The genome analysis and phylogenetic methods described in Methods sections "Reference guided assembly" and "Phylogeny" are appropriate and state-of-the-art. However, no information is given as to how and where strains were whole-genome sequenced. Also, the genomes of newly sequenced strains in this study should be uploaded to the EBI ENA and/or NCBI SRA, and the genome accession numbers of individual strains included in Table S1. The data of the RNA sequencing should also be uploaded to GEO (<https://www.ncbi.nlm.nih.gov/geo/>) and to the EBI ENA and/or NCBI SRA. This is to allow re-use of raw sequencing data (the same way the authors could re-use published genomes) and reproducibility of the results. In Supplementary Table 1 each strain should be accompanied by their genome accession numbers (i.e. run accession), both for published genomes and those sequenced in this study.

2. GWAS methodology. The Fst outlier analysis is not, to my knowledge, an established GWAS method to identify genetic variants associated with a phenotype. A more established GWAS method, such as Pyseer (<https://github.com/mgalardini/pyseer>) or this kmer GWAS (https://github.com/danny-wilson/kmer_pipeline), should have been used.

3. MIC distribution and ECOFF. Figure 1a. The distribution of metronidazole MICs in the presence of heme should be presented as an MIC distribution, as opposed to plotting individual MIC values sorted increasingly. For example, see EUCAST's metronidazole MIC distribution as an example: https://mic.eucast.org/search/show-registration/19320?back=https://mic.eucast.org/search/?search%255Bmethod%255D=mic%26search%255Bantibiotic%255D=126%26search%255Bspecies%255D=-1%26search%255Bdisk_content%255D=-1%26search%255Blimit%255D=50
Also, it would be important to establish if EUCAST's metronidazole ECOFF (2 ug/ml) matches or differs from the metronidazole MIC distribution (in the presence of heme) measured in this study.

Minor comments:

1. Abstract. In statements "PnimBG mutation was strongly associated with the Thr82Ile substitution conferring fluoroquinolone resistance" and "PnimBG also carried the Thr82Ile mutation" please indicate "Thr82Ile mutation in GyrA" as it reads as if the Thr82Ile mutation is carried in the nimB gene.

2. The quality and resolution of the phylogenetic trees presented in Figure 4 needs improvement. Also numbered or lettered panels are needed in this Figure. A phylogenetic tree of bigger size and resolution is needed, maybe included as a single figure.

Responses to Review Comments

We thank the reviewers for your time and efforts in reviewing our manuscript. Your collective comments have helped to improve the quality of this revised submission. Following the critiques, we conducted further experiments involving NimB enzyme assays, cellular assays, and Pyseer population analysis. As in the first submission, these experiments led to the same conclusions. Full details are described below in the point-by-point rebuttal. **NOTE:** Excel sheets were uploaded for: Source data; Supplementary Table S1. PDFs were uploaded for Supplementary Table S2 and S3.

Reviewer 1

1. Queries block A: Enzymological evaluation of NimB. The authors' description of NimB's role as nitroreductase is confusing because they alternately propose the transfer of two and of six electrons to the nitro group of metronidazole. In fact, the reduction of the nitro group to a comparably non-reactive amine requires the transfer of six electrons whereas the transfer of two electrons results in the formation of a toxic nitrosoradical. Indeed, already the original hypothesis as proposed by Leiros et al. 2004 was based on the IMO incorrect assumption that the formation of nitrosoradicals would have a protective effect. The authors show that the nitro groups in the respective nitro drugs (Figure 3) are reduced to their cognate amino groups. Thus, the authors should stick to this explanation dating back to the observations made by Carlier et al. 1997 (paper cited by the authors) and not confuse it with the contrasting hypothesis by Leiros et al. (see Discussion lines 394 and 395).

Response: Carlier hypothesis. We simplified our write-up removing the above speculations to be more consistent with Carlier's hypothesis, see 170-172 and 470-471.

We agree with the reviewer and work by Carlier et al.¹, suggesting that six electrons are required to reduce nitro group to an amine. Carlier¹ described the process as this: ***"The six-electron reducing process would result in a low steady state of the toxic form of 5-Ni, because the reaction is very rapid"***. Our original intent was to not confound the Carlier's hypothesis. Rather, it was intended to advance the hypothesis by speculating that the six electrons are derived from either FAD (i.e., FADH₂) and/or from heme. This considered that Nim is related to other heme flavoenzymes, such as Anf3², which are thought to conduct rapid electron transfers due to close positioning of heme and FAD.

Nim as a nitroreductase. We could not properly interpret the above comment, as it is unclear if the reviewer is concerned that we called Nim a "nitroreductase" instead of a "5-nitroimidazole reductase". In general, the nitroreductase family contain enzymes that are dependent on FAD/FMN and NAD(P)H to metabolize nitro-containing substrates³. Hence, we used the term "nitroreductase", because CdNimB reduces 4-nitrobenzoic acid and 2-nitroimidazole, in addition to 5-nitroimidazole; this represents a more comprehensive description of the enzyme based on data on the manuscript.

2. I do not understand how 10 μM of NimB can be possibly saturated with hemin at a hemin concentration of 4 μM (Figure 3d), i.e. at a 2.5-fold excess of NimB over hemin. Please explain!

Response: We agree. This can be explained as follows. NimB is a heme binding protein and as such during expression and purification BL21(DE3), the protein binds cellular heme. It is known

that a fraction of the hemoprotein molecules will be purified in a halo-state when expressed in *E. coli* (Fiege et al. 2018⁴). Thus, to confirm that heme is present in the native purified NimB we used a Hemin Assay Kit (Sigma MAK036) that colorimetrically measures low concentrations of heme based on peroxidase activity. The results in **Extended Figure 7b** shows both the purified WT and His55Ala mutant NimB proteins contain heme, but there was more heme in the WT samples. However, we view the data as semi-quantitative for the following reasons: detection of heme is based on the peroxidase enzyme use of heme/hemin as a cofactor. Because a fraction of the purified NimB has heme bound, then the peroxidase protein will likely have to compete with NimB for the cofactor. We do not know the heme dissociation constants for NimB and peroxidase proteins. However, these semi-quantitative results do clearly show that heme is present in both enzymes, but it is likely more abundant in WT enzyme. We indicated the limitation of the assay in **Extended Figure 7** legend. Noteworthy, as documented in the methods the enzyme prepared for heme binding did not have addition of heme to bacterial lysates. We also note that we tried to remove the heme using the methylethylketone method⁵, but this led to the enzyme being precipitated and unusable.

Extended Figure 7c also shows that suspensions of the WT and mutant protein are yellow, which can be expected for proteins that bind flavins^{6,7}. Nim proteins have been crystalized with flavin ligands e.g., *Bacteroides thetaiotaomicron* NimB (PDB 2FG9) with FAD and *Clostridium acetobutylicum* NimA/NimC (PDB 2IG6) with FMN.

3. To the best of my knowledge the nitroreductase assay as described in this paper and as shown in Figure 3f-h is novel. In light of this its description is fairly scant and lacks mention of necessary controls. Further, some of the details do not add up.

Response: We respectfully disagree that the assay is novel. In the original submission we indicated that assay was based on work on Anf3² a NADPH-dependent flavocytochrome that binds heme. We have now expanded the description of the assay in the methods section. In the below rebuttal points, we address concerns on the assay development. If the reviewer was referring to use of pyranose oxidase/catalase, this was used in the assay for Anf3² and it is used to remove residual oxygen. If the reviewer was referring the Bratton-Marshall assay, this method is well established and was successfully applied to assess nitroreductase activity against chloramphenicol^{8,9} and measurement of 2-aminoimidazole¹⁰.

4. Did the authors omit hemin, FAD and NADPH in any control experiments? This would be important for conferring credibility to the model and the assay. If the authors' model of NimB as a hemin-, FAD- and NADPH-dependent nitroreductase really applied, then none of these components could be omitted for aminoaromatic compounds to be formed. This has to be tested.

In our original experimentation, we conducted such experiments to establish the assay; however, we have now systematically repeated these experiments and the data is shown in **Extended Figure 8**. The reaction is dependent on NADPH as the electron donor. Because the purified enzyme contains some FAD and hemin (**Extended Figure 7**), there is some reaction that occurs when only NADPH is added; noteworthy, the enzymes prepared for nitroreductase assays had hemin added to bacterial culture and lysis buffer (see methods), for the purpose of increasing amount of properly folded and active protein, as a widely used approach⁴. With addition of heme or FAD, the reaction product increases. We hope this data now further validates the assay and NimB as an NADPH dependent enzyme that utilizes heme and FAD.

5. Why was the concentration of nitro compounds so high? 5 mM of metronidazole is a concentration several orders of magnitude higher than the concentrations encountered in vivo (MIC of metronidazole in resistant *C. difficile* 1-16 µg/ml, i.e. 6 to roughly 100 µM). Did the authors also test lower concentrations? Importantly, even at this high concentration the amount of amino-metronidazole formed is very small and only slightly above control level (Figure 3f). Since the reaction time was 2h and the concentration of NimB in the reaction mixture amounted to 10 µM, this indicates a minimal turnover of approximately 1 h⁻¹ when taking metronidazole as substrate. This is close to nil and cannot explain resistance to metronidazole. In this context it is also important to mention that 300 µM of NADPH as added to the reactions is only sufficient to allow the formation of 100 µM amino compounds, one NADPH molecule providing two electrons via a hydride transfer. It is unclear to me why the authors applied such a low amount of NADPH in their assay. Finally, please transform the values on the y-axis in Figure 3f into µM, the mere absorption value is not sufficiently informative!

Substrate (5 mM) concentration. During the original assay development (prior to first submission), we could not detect measurable amine when metronidazole was used at micromolar concentrations with the Bratton-Marshall assay. Metronidazole reduction produces a 5-aminoimidazole that is extremely chemically unstable^{11,12}, and we speculate this contribute to inability to measure the amine when micromolar concentrations of metronidazole was used as the substrate. This was also noted by Carlier et al.¹ for the 5-aminoimidazole of dimetridazole. However, 5 mM metronidazole gave a signal-to-noise of ~2 and allowed amine detection. We therefore adopted 5 mM as the standard concentration throughout, from which to achieve a good S/N and show that NimB reduces selected nitro-substrates.

Comparison of cellular and enzymatic conditions. The enzymatic conditions show that NimB converts metronidazole, 2-nitroimidazole, 4-nitrobenzoic acid, dimetridazole to amines and the reaction is enzyme dependent. However, further method development is needed to analyze enzyme kinetics for the enzyme. This is an enzyme from an anaerobe, and while we performed the enzyme reactions anaerobically, this is not the same as being in the cellular environment. However, it does not undermine our discovery regarding the enzyme's ability to metabolize nitro substrates. Our cellular data also supports the conclusion that production of the amines is NimB dependent. We note that Carlier et al.¹ only studied the NimA within cells and did not study it enzymatically; we are also not aware of reports where experiments showed the purified enzyme acts as a nitroreductase. Therefore, our findings are novel and represent a start toward conducting more advanced experimentation.

Enzyme turnover. We cannot say that the enzyme turnover for metronidazole is low, rather detection of the end-product poses considerable challenges due to its chemical instability. Even when we increased NADPH concentration to 3 mM the end-product detected was still low for metronidazole (see **Extended Figure 8**).

NADPH. We ran a concentration series for NADPH, and 3 mM was the lowest concentration giving maximum detection for metronidazole and nitroimidazole using the Bratton-Marshall assay. This is 10 times more than that previously used (0.3 mM) in the original submission. We repeated experiments with 3 mM of NADPH and reached the same conclusions as before.

Use of absorption as axis. It is not possible to change the scale to micromolar as the product for each substrate is different and standards are not available for each product (see **Figure 3f**). This plot differs from those in which nitrobenzoic acid and 2-nitroimidazole were the substrates

(see **Figures 3g-i**), because we could generate standard curves using 4-aminobenzoate or 2-aminoimidazole. The 5-aminoimidazole derivative of metronidazole is not commercially available. We sourced a 5-aminoimidazole analog of dimetrinidazole, during a different study, but GC-MS analysis indicated it had either degraded or was the incorrect material marketed by the only supplier offering the chemical.

6. Why was the amount of 2-nitroimidazole formed by the NimB H55A mutant so much lower when 2-aminoimidazole was quantified by Bratton-Marshall assay (Figure 3h) as compared to HPLC (Figure 3i)?

These experiments were repeated using 3 mM NADPH and new data is presented in Figures 3 h and i. We still see differences in relative quantities of amine detected by the LC-MS/MS and Bratton-Marshall assays, but both findings reached the same conclusion that His55Ala mutant is less effective than the WT enzyme under the same conditions. In the legend of figure 3, we now also state *“There are differences in relative amounts of 2-aminoimidazole quantified by the LC-MS/MS and Bratton-Marshall methods, but the results from both reached the same conclusion that the mutant is less effective in forming the amine product”*.

7. In fact, the supposed activity of NimB H55A amounts to 75% of the wildtype enzyme when the reaction product is measured by HPLC (Figure 3h) although this mutation is shown to abrogate metronidazole resistance as conferred by NimB (see Figure 3C). Taken into account this discrepancy, the R20291 Tn::nimB mutant complemented with nimB H55A (Figure 3C) would have been an important addition to the experiment as shown in Figure 3j.

We performed the above suggested experiment with R20291 Tn::nimB mutant complemented with nimB H55A. The data is now included in **Extended Figure 9**. It shows that complementation with His55Ala produced 2-aminoimidazole amounts similar to the uncomplemented strain bearing the empty vector, while the strain complemented with the WT nimB produced more end-product. The new data was done at Texas A&M University Chemistry core, since equipment at the previously used analytical core (Baylor College of Medicine) became broken. We specified in the revised version the origins of the LC-MS/MS data in the manuscript.

8. According to the data shown in Figure 3c, conversion of 2-nitroimidazole in cells harbouring the nimB H55A mutant should be equal to R0291 Δ nimB.

We attained this data with R20291-Tn::nimB showing that the amount of amine formed for the mutant was similar to R20291-Tn::nimB with the empty vector (see **Extended Figure 9**).

9. According to the high level of 2- aminoimidazole formation by NimB H55A as determined by HPLC, however, it should be close to the level of wildtype R20291.

In repeated experiments in which we increased NADPH to 3 mM, the His55Ala mutant enzyme consistently produced less 2-aminoimidazole than the WT enzyme. We conducted head-to-head comparisons of the WT and mutant enzymes with various substrates (**Figure 3f**). The data showed that in all cases the WT enzyme produced more amine product. Also see above for data in R20291-Tn::nimB.

10. Further also the MIC for 2-nitroimidazole in the presence of heme was strongly elevated in the metronidazole-susceptible strains CD196 and R20291 Δ nimB when nimB H55A was introduced via plasmid pRPF185 (Extended Data Figure 4). These issues show that 2-nitroimidazole is a poor substitute for metronidazole which is not surprising given that 2-nitroimidazoles have much higher reduction potentials than 5-nitroimidazoles, including metronidazole. They are more easily reduced than 5-nitroimidazoles and cause different effects. The 5-nitroimidazole dimetridazole as used by Carlier et al. 1997 would have been a much more suitable choice for these experiments.

Justification for using 2-nitroimidazole and comparison with Carlier's study. The purpose of using 2-nitroimidazole is because it offers a nitroimidazole analog that recapitulates heme-mediated resistance in cells and allowed us to measure cellular activity which can be quantified against a commercially available product (2-aminoimidazole). As described above, we could not routinely use dimetridazole, because there was no reliable standard (amine end-product). We also note that the 5-aminoimidazole are intrinsically unstable and becomes degraded in air, which would prevent its use in most analytical chemistry cores. **Data for dimetridazole is now presented in Figure 3f.**

We should also emphasize that Carlier et al.¹ appears to have also performed mass spec analysis under anaerobic conditions, which is a set-up that is not available. Carlier also described the amine end-product of dimetridazole as being unstable in air, which would make dimetridazole ineffective as a substrate for our cellular assays, where samples are sent for LC-MS/MS.

Variation in 2-nitroimidazole MICs. In our opinion, the data does not indicate that pnimBHis55Ala still confers resistance, but rather shows that biological variation of MIC testing. We performed additional MIC measurements with a new batch of 2-nitroimidazole and added this data to the plot with the prior reported data. Together, the data shows that complementation with pnimB-WT always leads to resistance at levels comparable to R20291-WT, in contrast to the pnimBHis55Ala or the empty vector. The replotted graphs, per journal requirement, now shows all datapoints, which supports the above conclusion.

Based on the critique, we also improved clarity for readers that MICs of metronidazole against R20291 Tn::nimB tend to be slightly higher than MICs against R20291 Δ nimB, because we suspect that the mutant enzyme in the *Tn* mutant might have some activity since the insertion only disrupted the C terminal region. This is now explained in the manuscript (lines 180-183).

11. It would be interesting to see overexpression of nimB plotted against the degree of metronidazole resistance. From the data I could muster in the manuscript no correlation between the degree of resistance and nimB mRNA levels can be inferred. Strain 234

68, e.g., displays an almost 20-fold overexpression of nimB in the presence of heme as compared to the parental 23475 strain but is only resistant to a metronidazole conc. of 1 μ g/ml (Figure 2d). Strain 70/76, displaying only an about 5-fold overexpression of nimB as compared to susceptible CD196, is resistant to 8 μ g/ml of metronidazole (see Wu et al., 2021 from the same lab). The same goes for the couple 17/27 and 25603 (Extended Figure 5). Strain 17/27 has much lower expression of nimB than 25603 but is far more resistant. This suggests that NimB levels do not translate into the level of metronidazole resistance as such which complicates the notion of NimB acting directly as a nitroreductase. Higher levels of nitroreductase should lead to higher levels of detoxified metronidazole. This aspect should be carefully discussed by the authors.

Response: In **Extended Figure 4b**, we showed that resistance vs susceptible strains can be partitioned based on the qPCR C_T values. However, more strains would need to be tested to

devise a C_T value to pinpoint resistance versus susceptible strains by molecular diagnostics. Use of the C_T value is common for molecular diagnostics. We also recognize there has been a long-standing debate on expression of *nim* genes and detection of resistance; studies have shown a link between expression and resistance, while others have not^{13,14}. However, at the proteomic and metabolomic levels there are several factors to consider, across biological distinct strains. This includes (a) kinetics of uptake of heme and metronidazole, (b) cellular availability of heme and flavin cofactors for use by Nim versus other flavin/hemo enzymes; (c) cellular dynamics of inactivation of nitro-group to amine versus its activation to free radicals i.e., dynamics between Nim enzyme versus other enzymes (e.g., PFOR, xanthine dehydrogenase etc.). We believe these questions require much more intensive research, which was beyond the scope of this discovery-oriented study.

12. Contrary to the authors' claim (lines 335-337), the inhibition of *nimB* expression in strains 17/27 and 26503 (Extended Figure 5d), both having elevated *nimB* mRNA levels but nor mutation in the promoter region, does not result in the restoration of metronidazole susceptibility to an extent comparable to the strains shown in Figure 2g and h. Surprisingly, antisense mRNA has no effect at all and CrispR only partially restores susceptibility in 17/27. Please explain!

Response: We repeated experiments and reached similar conclusions (**Extended Figure 11**); we combined the old and new data for a total of six biological replicates. First to clarify, these antisense experiments on 17/27 and 26503 involved use of the ATc inducible pMSPT vector and not the xylose-inducible CrispR system. The previous submission showed both strains displayed resistance with the empty vector (with or without ATc). In strains with the antisense vector resistance is attenuated when the antisense is induced with ATc. In repeated experiments 17/27 still did not become as susceptible as 26503 in the presence of ATc. There are multiple factors to consider: (a) the two strains are different genetic backgrounds, with strain 26503 being a ribotype 027, like all strains in Figure 2 (i.e., all RT027 backgrounds); strain 17/27 is a RT001 genetic background. (b) 17/27 may contain additional *NimB* independent resistance as well. We have amended the write-up taking these factors into consideration.

13. The growth experiment with strains 23468 and 23475 as discussed in the Discussion (lines 381-383) is not sufficient to show that *nimB* overexpression has no fitness costs. The experiment was performed in BHI medium without hemin although hemin is required to observe a phenotype of *nimB* overexpression. I think the experiment should be presented in the Results section and not in the Discussion.

Response: We re-performed the experiment with and without hemin and see no effect on fitness. We have added the data to the results section.

14. The data shown in Extended Figure 7 and discussed in the Discussion lines 399-403 should also be transferred to the Results. The results with *NimA* from *B. fragilis* are interesting and mirror our own observations. However, as a word of caution: resistance to metronidazole in Bf 638R pIP417 (*nimA*) is also clearly increased when the cells are plated on BHI plates with extra FeCl₃ (our own as yet unpublished results) which is not the case in *C. difficile*, at least according to the authors' previous results (Wu et al., 2021). It is therefore uncertain if the mechanism is the same as observed in *C. difficile*.

Response: We thank the reviewer for sharing unpublished data on Bf 638R and to know our data mirrors the reviewer's data. Regarding the increase in metronidazole MICs in FeCl₃, it may be possible that for Bf the added iron affects flavin cofactors.

As requested, we have moved data on *B. fragilis* from the Discussion to the Results section of the paper. This is now Extended Data Figure 10.

15. To sum up, this paper constitutes an important milestone in *C. difficile* research and definitely deserves to be swiftly published. The part on the presumed nitroreductase activity, however, requires much more attention and care before publication. In order to avoid an unduly delay of publication of the manuscript, I would therefore suggest to shorten this part of the manuscript and expand on the issue in a follow-up paper.

Response: We thank the reviewer for the very positive comments on the manuscript. Based on the above criticisms of the enzyme experiments, we conducted additional experiments that improved the quality of the manuscript. These results, along with the cellular data, support the description of the enzyme as a nitroreductase. Together these findings advance the limited knowledge of Nim protein biochemistry, combining our cellular and enzymatic *in vitro* assays. As far as we are aware, Nim proteins have not been successfully tested in enzyme assays for nitroreductase activity; this may be due to a lack of knowledge of heme as a cofactor. However, the results in this study provide an invaluable starting point for future experiments to understand their molecular mechanisms (the subject of our future work).

Reviewer 2

16. The manuscript "Decoding a cryptic mechanism of metronidazole resistance among globally disseminated fluoroquinolone-resistant *C. difficile*" combines a myriad of methods to dissect a mechanism of resistance to metronidazole that is associated with a *gryA* mutation that was previously shown to facilitate the global dissemination of epidemic Cd strains. The conclusions are in most cases well supported by the data and have multiple theoretical and pragmatical applications.

This reviewer believes that the authors confuse causality with association at some points along the text but this can be addressed by changing the wording.

Response: We thank the reviewer for overall positive comments and have addressed concerns regarding causality vs association at different points in the manuscript.

17. Abstract: indicate %id of *CdNimB* to other *nim* genes from other species. Include MICs and change folds to appraise the magnitude of the phenotype.

Response: We modified the abstract indicating *CdNimB* is a pyridoxamine 5'-phosphate oxidase family protein related to Nim proteins that confer resistance to nitromidazoles. Recognizing there are multiple strains and aspects of the study, we limited MICs to strain R20291, which is well-known in the research community and is a representative epidemic RT027 strain. Information of protein relatedness to other Nim proteins is stated in the results section under "*Identification of 5-nitroimidazole reductase (CdNimB) as a mechanism for C. difficile heme-dependent metronidazole resistance.*"

18. Introduction: l47: belongs to "MLST Clade 2". There are different grouping methods, in this work only MLST is used and this should be indicated for the sake of clarity.

Response: In this work we used both ribotyping and phylogenetic classification. We modified the Introduction to state *"in this study, strains are classified based on their ribotype and/or phylogenetic clade, unless otherwise specified"*.

19. I49. also in Latin America.

Response: We agree and made this change.

20. I88. Please elaborate the mechanism of action of metronidazole so that the reader can later understand the transcriptomics responses.

Response: We agree and made this change.

21. I90. "revealing its role in the global transmission" This is speculative, what the authors can claim with the current experimental design is association/linkage, not causality. It would be advisable to present metronidazole consumption figures or usage data to better understand the strength of the selective pressure in the countries where the isolates were recovered.

Response: We agree and now use the term "association" instead of "role". However, data on global consumption of metronidazole is unavailable from where isolates were recovered.

22. I99. To ultimately confirm co-selection, additional experiments in, say, a human gut model, are required. This can be discussed.

Response: We have clarified the terminology in the updated manuscript and removed the term co-selection. We hypothesize that the *nimB* promoter and *gyrA* mutations are subject to positive, directional selection. We tested this hypothesis with homoplasmy analyses, the results of which were consistent with positive selection acting on the resistance alleles and maintaining both mutations in natural populations. We agree with the reviewer about the role of further experimentation to investigate the relationship between these resistances and have added this to the discussion (lines 465-467).

23. I153. please add "in *C. difficile*" before "since".

Response: We have made this change.

24. I159. please add "plasmid" before "plasmid".

Response: We were unsure what this comment means; we now indicate that *nimB* was plasmid-borne in the complementation experiment.

25. I169. for consistency, please indicate ST and MLST clade of isolates 23475 and 23468.

Response: We have now included that strains are cgMLST 20612 from Enterobase and their ribotype as RT014.

26. I232 and 234. Homologous refers to common ancestry, here you mean similar 3D structure.

Response: We believe we have used the term in the right context, relating to protein homology. Use of these terms are also compatible with the protein science literature. However, we now use the term “structurally homologous” in the first sentence (formerly line 232) and the next sentence retained the term “homologous” as it is contextual.

27. I245. please indicate method.

Response: This has now been done.

28. I339-340. Perhaps knowledge from other bacteria can shed light onto this alternative mechanism, please consider whether this can be discussed.

Response: We have now cited past work on other mechanism in *C. difficile*.

29. Discussion: I352. Indicating? To me the paper can not confirm this claim.

Response: This statement has been modified.

30. I361. It is a bit reckless to conclude "that strains carrying both mutations spread more rapidly". Our current knowledge on the global population of Cd is highly biased towards a limited number of lineages linked to some phenotypes.

Response: Our statement regarding the rapid spread of epidemic *C. difficile* is consistent with the existing literature suggesting that these strains had spread rapidly around the world¹⁵⁻¹⁷. In our study, we also examined the “global dataset” of strains from He et al.¹⁶ which is well cited as providing strong support for rapid continental spread of epidemic RT027 strains. Our analysis showed that all but one of these strains had PnimB^G, indicating that the conclusions drawn in the He et al paper also apply to our study. Our study however updates the paradigm for fluoroquinolone resistant *C. difficile*, as these strains are also mostly likely to be resistant to metronidazole due to PnimB^G. We do agree with the reviewer that current knowledge is highly biased because strains and lineages have not been collected and studied globally.

31. I364, 365, 368, 373. Clade with capital C.

Response: Clade has been capitalized throughout.

32. I404. In the abstract the gene is called nimB, here only nim. What’s the correct nomenclature and why?

Response: We corrected the gene from *nim* to *nimB*. *nimB* is the correct nomenclature of the gene in *C. difficile*, while *nim* is the general description of the gene among bacteria; in bacteria there are different *nim* genes (*nimA*, *nimB* etc).

33. Congratulation on such a great paper, very exhaustive experimental design and a great contribution to the field.

Response: Many thanks for your constructive comments that have improved the manuscript's quality.

Reviewer 3

34. In this study, Olaitan and colleagues characterise a new mechanism of metronidazole resistance in *Clostridioides difficile*. This is an excellent multi-disciplinary study where authors provide extensive genetic, biochemical and evolutionary evidence of the role of over-expression of NimB in metronidazole resistance. Find below my major and minor comments:

Response: We thank the reviewer for overall positive comments.

35. Data availability. The genome analysis and phylogenetic methods described in Methods sections “Reference guided assembly” and “Phylogeny” are appropriate and state-of-the-art. However, no information is given as to how and where strains were whole-genome sequenced. Also, the genomes of newly sequenced strains in this study should be uploaded to the EBI ENA and/or NCBI SRA, and the genome accession numbers of individual strains included in Table S1. The data of the RNA sequencing should also be uploaded to GEO (<https://www.ncbi.nlm.nih.gov/geo/>) and to the EBI ENA and/or NCBI SRA. This is to allow re-use of raw sequencing data (the same way the authors could re-use published genomes) and reproducibility of the results. In Supplementary Table 1 each strain should be accompanied by their genome accession numbers (i.e. run accession), both for published genomes and those sequenced in this study.

Response: We agree; in the revised submission we specified where the genomes were sequenced, and the method used. We have deposited the genomes in NCBI and modified the **Supplementary Table S1** with their accession numbers. The raw RNAseq files have also been deposited in NCBI and the accession number included in the manuscript.

36. GWAS methodology. The Fst outlier analysis is not, to my knowledge, an established GWAS method to identify genetic variants associated with a phenotype. A more established GWAS method, such as Pyseer (<https://github.com/mgalardini/pyseer>) or this kmer GWAS (https://github.com/danny-wilson/kmer_pipeline), should have been used.

Response: We have added an analysis using Pyseer, which gave identical results to the FST outlier analysis. We also cite a published study using FST outlier analysis to link genetic variants with specific phenotypes.

37. MIC distribution and ECOFF. Figure 1a. The distribution of metronidazole MICs in the presence of heme should be presented as an MIC distribution, as opposed to plotting individual MIC values sorted increasingly. For example, see EUCAST’s metronidazole MIC distribution as an example: https://mic.eucast.org/search/show-registration/19320?back=https://mic.eucast.org/search/?search%255Bmethod%255D=mic%26search%255Bantibiotic%255D=126%26search%255Bspecies%255D=-1%26search%255Bdisk_content%255D=-1%26search%255Blimit%255D=50

Response: We recognize the importance of the EUCAST plot to show MIC distribution and its usefulness for tracking epidemiological shifts in MICs. We now show this in **Extended Data Figure 1a-b**. We retained the style in Fig 1a as it is designed to show the effect of heme on MICs for each strain tested in **Supplementary Table S1**, with the inset focusing on historic and

epidemic ribotype 027 strains. Each MIC is also given in the supplementary table, allowing future users to evaluate the genomes alongside MIC data.

38. Also, it would be important to establish if EUCAST's metronidazole ECOFF (2 ug/ml) matches or differs from the metronidazole MIC distribution (in the presence of heme) measured in this study.

Response: In the new **Extended Data Figure 1**, we compared MIC distributions with and without heme showing heme shifts MICs for the resistant strains toward the ECOFF (2 $\mu\text{g/ml}$). However, we previously reported that an MIC of 1 $\mu\text{g/ml}$ was more associated with clinical failure¹⁸. Some resistant strains carrying *PnimB^G* have a metronidazole MIC of 1 $\mu\text{g/ml}$ in heme and 0.25 $\mu\text{g/ml}$ without heme, for e.g., strain 23468 in Figure 2. Such strains would be missed as susceptible by the current ECOFF. We hope the present study could help to revisit the ECOFF for metronidazole and inspire genetic test of *PnimB^G* by clinical labs.

39. Abstract. In statements "PnimBG mutation was strongly associated with the Thr82Ile substitution conferring fluoroquinolone resistance" and "PnimBG also carried the Thr82Ile mutation" please indicate "Thr82Ile mutation in GyrA" as it reads as if the Thr82Ile mutation is carried in the nimB gene.

Response. This has now been done.

40. The quality and resolution of the phylogenetic trees presented in Figure 4 needs improvement. Also numbered or lettered panels are needed in this Figure. A phylogenetic tree of bigger size and resolution is needed, maybe included as a single figure.

Response. This has now been done.

1. Carlier, J.P., Sellier, N., Rager, M.N. & Reyssset, G. Metabolism of a 5-nitroimidazole in susceptible and resistant isogenic strains of *Bacteroides fragilis*. *Antimicrob Agents Chemother* **41**, 1495-1499 (1997).
2. Varghese, F., *et al.* A low-potential terminal oxidase associated with the iron-only nitrogenase from the nitrogen-fixing bacterium *Azotobacter vinelandii*. *J Biol Chem* **294**, 9367-9376 (2019).
3. Koder, R.L. & Miller, A.F. Overexpression, isotopic labeling, and spectral characterization of *Enterobacter cloacae* nitroreductase. *Protein Expr Purif* **13**, 53-60 (1998).
4. Fiege, K., Querebillo, C.J., Hildebrandt, P. & Frankenberg-Dinkel, N. Improved Method for the Incorporation of Heme Cofactors into Recombinant Proteins Using *Escherichia coli* Nissle 1917. *Biochemistry* **57**, 2747-2755 (2018).
5. Teale, F.W. Cleavage of the haem-protein link by acid methylethylketone. *Biochim Biophys Acta* **35**, 543 (1959).
6. Watanabe, M., Nishino, T., Takio, K., Sofuni, T. & Nohmi, T. Purification and characterization of wild-type and mutant "classical" nitroreductases of *Salmonella typhimurium*. L33R mutation greatly diminishes binding of FMN to the nitroreductase of *S. typhimurium*. *J Biol Chem* **273**, 23922-23928 (1998).
7. Valentino, H., *et al.* Structural and Biochemical Characterization of the Flavin-Dependent Siderophore-Interacting Protein from *Acinetobacter baumannii*. *ACS Omega* **6**, 18537-18547 (2021).

8. Smith, A.L., *et al.* Chloramphenicol is a substrate for a novel nitroreductase pathway in *Haemophilus influenzae*. *Antimicrob Agents Chemother* **51**, 2820-2829 (2007).
9. Mallowney, M.W., *et al.* Functional and Structural Characterization of Diverse NfsB Chloramphenicol Reductase Enzymes from Human Pathogens. *Microbiol Spectr* **10**, e0013922 (2022).
10. Seki, Y., Nakamura, T. & Okami, Y. Accumulation of 2-aminoimidazole by *Streptomyces eurocidicus*. *J Biochem* **67**, 389-396 (1970).
11. Lichitsky, B.V., Komogortsev, A.N., Dudinov, A.A. & Krayushkin, M.M. Three-component condensation of 5-aminoimidazole derivatives with aldehydes and Meldrum's acid. Synthesis of 3,4,6,7-tetrahydroimidazo[4,5-b]pyridin-5-ones. *Russian Chemical Bulletin* **61**, 1591-1595 (2012).
12. Hofmann, K. *Imidazole and its Derivatives, Part 1*, (Interscience Publishers, New York, 1953).
13. Leitsch, D., Soki, J., Kolarich, D., Urban, E. & Nagy, E. A study on Nim expression in *Bacteroides fragilis*. *Microbiology (Reading)* **160**, 616-622 (2014).
14. Kupc, M., Paunkov, A., Strasser, D., Soki, J. & Leitsch, D. Initial expression levels of *nimA* are decisive for protection against metronidazole in *Bacteroides fragilis*. *Anaerobe* **77**, 102630 (2022).
15. Valiente, E., Cairns, M.D. & Wren, B.W. The *Clostridium difficile* PCR ribotype 027 lineage: a pathogen on the move. *Clin Microbiol Infect* **20**, 396-404 (2014).
16. He, M., *et al.* Emergence and global spread of epidemic healthcare-associated *Clostridium difficile*. *Nat Genet* **45**, 109-113 (2013).
17. Clements, A.C., Magalhaes, R.J., Tatem, A.J., Paterson, D.L. & Riley, T.V. *Clostridium difficile* PCR ribotype 027: assessing the risks of further worldwide spread. *Lancet Infect Dis* **10**, 395-404 (2010).
18. Gonzales-Luna, A.J., *et al.* Reduced Susceptibility to Metronidazole Is Associated With Initial Clinical Failure in *Clostridioides difficile* Infection. *Open Forum Infect Dis* **8**, ofab365 (2021).

Reviewer #1 (Remarks to the Author):

The authors have truly invested lots of time and effort to answer my queries and I agree that the manuscript has been greatly improved by the modifications. I therefore advise publication of the manuscript although I still disagree that the protection conferred against metronidazole by NimB is due to the enzyme's nitroreductase activity. Their enzyme assay shows that the rate of reduction must be really low and the K_m for NADPH extremely high and the lack of correlation between resistance and NimB expression levels is still puzzling. However, the authors provide enough and technically sound data for their hypothesis to be published and to be discussed within the research community. I hope that the authors continue their studies on NimB and can discover the canonical function of the enzyme. Congratulations!

Response to critique

Reviewer #1 (Remarks to the Author):

The authors have truly invested lots of time and effort to answer my queries and I agree that the manuscript has been greatly improved by the modifications. I therefore advise publication of the manuscript although I still disagree that the protection conferred against metronidazole by NimB is due to the enzyme's nitroreductase activity. Their enzyme assay shows that the rate of reduction must be really low and the K_m for NADPH extremely high and the lack of correlation between resistance and NimB expression levels is still puzzling. However, the authors provide enough and technically sound data for their hypothesis to be published and to be discussed within the research community. I hope that the authors continue their studies on NimB and can discover the canonical function of the enzyme. Congratulations!

Response: We thank the reviewer for pointing out that the manuscript has been greatly improved through the additional work. We respect the author disagreement about the protection conferred by NimB, based on the enzyme assay findings. We recognize that our findings moves the field forward for NimB. As such we agree with the statement by the reviewer that *“the authors provide enough and technically sound data for their hypothesis to be published and to be discussed within the research community”*. To address this critique further, the following statement is found in the discussion:

“Based on the homodimeric homology model from Anf3 structure (Fig. 3b), and biochemistry of heme flavoenzymes, we speculate CdNimB may reduce the nitroimidazole nitro-groups with electrons from heme or flavin cofactors in the homodimeric protein. Further research is required to test these hypotheses in more optimal biochemical and biophysical experiments to establish the molecular mechanism(s) through which Nim proteins reduce nitroimidazoles. As our experiments used millimolar substrate concentrations, these further studies should include an analysis of the enzyme's kinetic parameters. This will provide a more accurate assessment of the enzyme's catalytic efficiency in reducing physiologic, micromolar concentrations of metronidazole.”